# Private GANs, Revisited[*]

**Alex Bie**[†]                          *yabie@uwaterloo.ca*
*University of Waterloo*

**Gautam Kamath**                          *g@csail.mit.edu*
*University of Waterloo*

**Guojun Zhang**                     *guojun.zhang@huawei.com*
*Huawei Noah's Ark Lab*

*Reviewed on OpenReview:* *https://openreview.net/forum?id=9sVCIngrhP*

## Abstract

We show that the canonical approach for training differentially private GANs – updating the discriminator with differentially private stochastic gradient descent (DPSGD) – can yield significantly improved results after modifications to training. Specifically, we propose that existing instantiations of this approach neglect to consider how adding noise *only to discriminator updates* inhibits discriminator training, disrupting the balance between the generator and discriminator necessary for successful GAN training. We show that a simple fix – taking more discriminator steps between generator steps – restores parity between the generator and discriminator and improves results.

Additionally, with the goal of restoring parity, we experiment with other modifications – namely, large batch sizes and adaptive discriminator update frequency – to improve discriminator training and see further improvements in generation quality. Our results demonstrate that on standard image synthesis benchmarks, DPSGD outperforms all alternative GAN privatization schemes. Code: https://github.com/alexbie98/dpgan-revisit.

## 1 Introduction

Differential privacy (DP) (Dwork et al., 2006b) has emerged as a compelling approach for training machine learning models on sensitive data. However, incorporating DP requires changes to the training process. Notably, it prevents the modeller from working directly with sensitive data, complicating debugging and exploration. Furthermore, upon exhausting their allocated privacy budget, the modeller is restricted from interacting with sensitive data. One approach to alleviate these issues is by producing *differentially private synthetic data*, which can be plugged directly into existing machine learning pipelines, without further concern for privacy.

Towards generating high-dimensional, complex data (such as images), a line of work has examined privatizing generative adversarial networks (GANs) (Goodfellow et al., 2014) to produce DP synthetic data. Initial efforts proposed to use differentially private stochastic gradient descent (DPSGD) (Abadi et al., 2016) as a drop-in replacement for SGD to update the GAN discriminator – an approach referred to as *DPGAN* (Xie et al., 2018; Beaulieu-Jones et al., 2019; Torkzadehmahani et al., 2019).

However, follow-up work (Jordon et al., 2019; Long et al., 2021; Chen et al., 2020; Wang et al., 2021) departs from this approach: they propose alternative privatization schemes for GANs, and report significant improvements over the DPGAN baseline. Other methods for generating DP synthetic data diverge from GAN-based architectures, yielding improvements to utility in most cases (Table 2). This raises the question of whether GANs are suitable for DP training, or if bespoke architectures are required for DP data generation.

---

[*]Authors GK and GZ are listed in alphabetical order.
[†]Work performed in part while interning at Huawei.

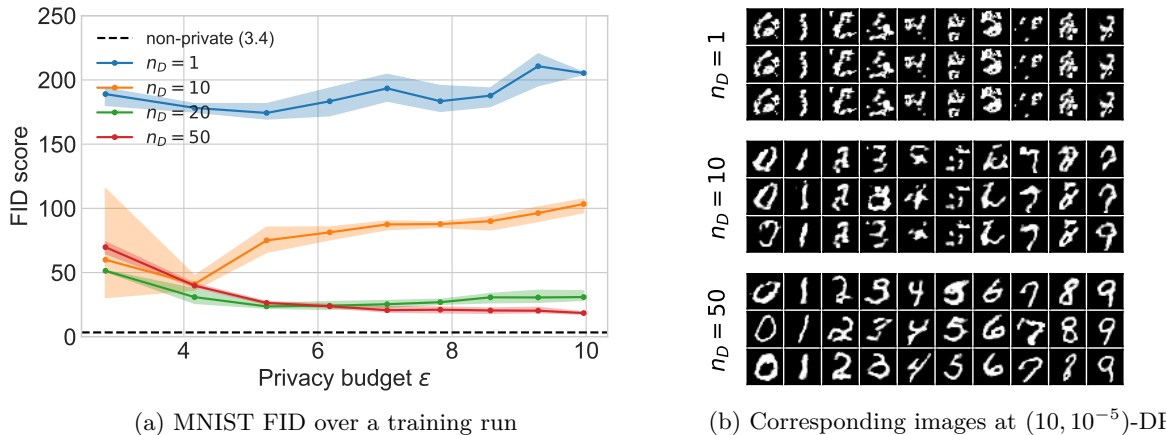

(a) MNIST FID over a training run    (b) Corresponding images at $(10, 10^{-5})$-DP

Figure 1: DPGAN results on MNIST synthesis at $(10, 10^{-5})$-DP. **(a)** We run 3 seeds, plotting mean, min, and max FID along the runs. We find that increasing $n_{\mathcal{D}}$, the number of discriminator steps taken between generator steps, significantly improves image synthesis. Increasing $n_{\mathcal{D}} = 1 \to n_{\mathcal{D}} = 50$ improves FID from $205.3 \pm 0.9 \to 18.5 \pm 0.9$. **(b)** Corresponding synthesized images (each are trained with the same privacy budget). We observe that large $n_{\mathcal{D}}$ improves visual quality, and low $n_{\mathcal{D}}$ leads to mode collapse.

| | | MNIST | | FashionMNIST | | CelebA-Gender | |
|---|---|---|---|---|---|---|---|
| Privacy $\varepsilon$ | Method | FID | Acc.(%) | FID | Acc.(%) | FID | Acc.(%) |
| $\varepsilon = \infty$ | Real Data | 1.0 | 99.2 | 1.5 | 92.5 | 1.1 | 96.6 |
| | GAN | $3.4 \pm 0.1$ | $97.0 \pm 0.1$ | $16.5 \pm 1.7$ | $79.5 \pm 0.8$ | $30.0 \pm 1.6$ | $92.0 \pm 0.4$ |
| $\varepsilon = 10$ | Best Private GAN | 61.34 | 80.92 | 131.34 | 70.61 | - | 70.72 |
| | DPGAN | 179.16 | 80.11 | 243.80 | 60.98 | - | 54.09 |
| $\varepsilon = 9.32$ | Our DPGAN | $12.8 \pm 0.3$ | $95.1 \pm 0.1$ | $62.3 \pm 8.7$ | $74.7 \pm 0.4$ | $170.8 \pm 20.3$ | $82.4 \pm 4.4$ |

Table 1: A summary of our results compared to results reported in previous work on private GANs. For our results, we run 3 seeds and report mean $\pm$ std. *Acc.(%)* refers to downstream classification accuracy of CNN models trained with generated data. The middle two rows are a composite of the best results reported in the literature for DPGAN and alternative GAN privatization schemes (*"Best Private GAN"*); see Tables 2 and 3 for correspondences. Here we use Gopi et al. (2021) privacy accounting for our results. We find significant improvement over all previous GAN-based methods for DP image synthesis.

**Our contributions.** We show that DPGANs give far better utility than previously demonstrated, and compete with or outperform almost all other methods for DP image synthesis.[1]

Hence, we conclude that previously demonstrated deficiencies of DPGANs should not be attributed to inherent limitations of the framework, but rather, training issues. Specifically, we propose that the *asymmetric noise addition* in DPGANs (adding noise to discriminator updates only) inhibits discriminator training while leaving generator training untouched, disrupting the balance necessary for successful GAN training. Indeed, the seminal study of Goodfellow et al. (2014) points to the challenge of "synchronizing the discriminator with the generator" in GAN training, suggesting that, "$\mathcal{G}$ must not be trained too much without updating $\mathcal{D}$, in order to avoid 'the Helvetica scenario' [*mode collapse*]". Prior DPGAN implementations in the literature do not take this into consideration in the process of porting over non-private GAN training recipes.

We propose that taking more discriminator steps between generator updates addresses the imbalance introduced by noise. With this change, DPGANs improve significantly (see Figure 1 and Table 1). Furthermore, we show this perspective on DPGAN training ("restoring parity to a discriminator weakened by DP noise") can be applied to improve training. We make other modifications to discriminator training – larger batch

---
[1]A notable exception is diffusion models, discussed further in Section 2.

sizes and adaptive discriminator update frequency – to improve discriminator training and further improve upon the aforementioned results. In summary, we make the following contributions:

- We find that taking more discriminator steps between generator steps significantly improves DP-GANs. Contrary to previous results in the literature, DPGANs outperform alternative GAN privatization schemes.

- We present empirical findings towards understanding why more frequent discriminator steps help. We propose an explanation based on *asymmetric noise addition* for why vanilla DPGANs do not perform well, and why taking more frequent discriminator steps helps.

- We employ our explanation as a principle for designing better private GAN training recipes – incorporating larger batch sizes and adaptive discriminator update frequency – and indeed are able to improve over the aforementioned results.

## 2 Related work

**Private GANs.** The baseline DPGAN that employs a DPSGD-trained discriminator was introduced in Xie et al. (2018), and studied in follow-up work of Torkzadehmahani et al. (2019); Beaulieu-Jones et al. (2019). Despite significant interest in the approach ($\approx$400 citations at time of writing), we were unable to find studies that explore the modifications we perform or uncover similar principles for improving DPGAN training. We note that the number of discriminator steps taken per generator step, $n_{\mathcal{D}}$, appears as a hyperparameter in the framework outlined by the seminal study of Goodfellow et al. (2014), and in follow-up work such as WGAN (Arjovsky et al., 2017). Xie et al. (2018) privatizes WGAN, adopting its imbalanced stepping strategy of $n_{\mathcal{D}} = 5$, however makes no mention of the importance of the parameter (along with Torkzadehmahani et al. (2019), which uses $n_{\mathcal{D}} = 1$). As we show in Figure 1a, ensuring that $n_{\mathcal{D}}$ lies within a critical range (as determined by DPSGD hyperparameters) is key to adapting a GAN training recipe to DP; selection of $n_{\mathcal{D}}$ is the difference between state-of-the-art-competitive performance and something that is entirely not working.[2]

As a consequence, subsequent work has departed from DPGANs, examining alternative privatization schemes for GANs (Jordon et al., 2019; Long et al., 2021; Chen et al., 2020; Wang et al., 2021). Broadly speaking, these approaches employ subsample-and-aggregate (Nissim et al., 2007) via the PATE approach (Papernot et al., 2017), dividing the data into $\geq$ 1K disjoint partitions and training teacher discriminators separately on each one. Our work shows that these privatization schemes are outperformed by DPSGD.

**DP generative models.** Other generative modelling frameworks have been applied to generate DP synthetic data: VAEs (Chen et al., 2018), maximum mean discrepancy (Harder et al., 2021; Vinaroz et al., 2022; Harder et al., 2022), Sinkhorn divergences (Cao et al., 2021), normalizing flows (Waites & Cummings, 2021), and diffusion models (Dockhorn et al., 2022). In a different vein, Chen et al. (2022) avoids learning a generative model, and instead generates a coreset of examples ($\approx$ 20 per class) for the purpose of training a classifier. These approaches fall into two camps: applications of DPSGD to existing, highly-performant generative models; or custom approaches designed specifically for privacy which fall short of GANs when evaluated at their non-private limits ($\varepsilon \to \infty$).

**Concurrent work on DP diffusion models.** Simultaneous and independent work by Dockhorn et al. (2022) is the first to investigate DP training of diffusion models. They achieve impressive state-of-the-art results for DP image synthesis in a variety of settings, in particular, outperforming our results for DPGANs reported in this paper. We consider our results to still be of significant interest to the community, as we challenge the conventional wisdom regarding deficiencies of DPGANs, showing that they give much better utility than previously thought. Indeed, GANs are still one of the most popular and well-studied generative models, and consequently, there are many cases where one would prefer a GAN over an alternative approach. By revisiting several of the design choices in DPGANs, we give guidance on how to seamlessly introduce

---

[2]For further discussion on the role of hyperparameters in DP machine learning, see Appendix F.

differential privacy into such pipelines. Furthermore, both our work and the work of Dockhorn et al. (2022) are aligned in supporting a broader message: training conventional machine learning architectures with DPSGD frequently achieves state-of-the-art results under differential privacy. Indeed, both our results and theirs outperform almost all custom methods designed for DP image synthesis. This reaffirms a similar message recently demonstrated in other private ML settings, including image classification (De et al., 2022) and NLP (Li et al., 2022; Yu et al., 2022).

## 3 Preliminaries

Our goal is to train a generative model on sensitive data that is safe to release, that is, it does not leak the secrets of individuals in the training dataset. We do this by ensuring the training algorithm $\mathcal{A}$ – which takes as input the sensitive dataset $D$ and returns the parameters of a trained (generative) model $\theta$ – satisfies differential privacy.

**Definition 1** (Differential Privacy, Dwork et al. 2006b)**.** A randomized algorithm $\mathcal{A} : \mathcal{U} \to \Theta$ is $(\varepsilon, \delta)$-*differentially private* if for every pair of neighbouring datasets $D, D' \in \mathcal{U}$, we have

$$\mathbb{P}\{\mathcal{A}(D) \in S\} \leq \exp(\varepsilon) \cdot \mathbb{P}\{\mathcal{A}(D') \in S\} + \delta \qquad \text{for all (measurable) } S \subseteq \Theta.$$

In this work, we adopt the add/remove definition of DP, and say two datasets $D$ and $D'$ are neighbouring if they differ in at most one entry, that is, $D = D' \cup \{x\}$ or $D' = D \cup \{x\}$.

One convenient property of DP is *closure under post-processing*, which says that further outputs computed from the output of a DP algorithm (without accessing private data by any other means) are safe to release, satisfying the same DP guarantees as the original outputs. In our case, this means that interacting with a privatized model (e.g., using it to compute gradients on non-sensitive data, generate samples) does not lead to any further privacy violation.

**DPSGD.** A gradient-based learning algorithm can be privatized by employing *differentially private stochastic gradient descent* (DPSGD) (Song et al., 2013; Bassily et al., 2014; Abadi et al., 2016) as a drop-in replacement for SGD. DPSGD involves clipping per-example gradients and adding Gaussian noise to their sum, which effectively bounds and masks the contribution of any individual point to the final model parameters. Privacy analysis of DPSGD follows from several classic tools in the DP toolbox: Gaussian mechanism, privacy amplification by subsampling, and composition (Dwork et al., 2006a; Dwork & Roth, 2014; Abadi et al., 2016; Wang et al., 2019).

In our work, we use two different privacy accounting methods for DPSGD: (a) the classical approach of Mironov et al. (2019), implemented in Opacus (Yousefpour et al., 2021), and (b) the recent exact privacy accounting of Gopi et al. (2021). By default, we use the former technique for a closer direct comparison with prior works (though we note that some prior works use even looser accounting techniques). However, the latter technique gives tighter bounds on the true privacy loss, and for all practical purposes, is the preferred method of privacy accounting. We use Gopi et al. (2021) accounting only where indicated in Tables 1, 2, and 3.

**DPGANs.** Algorithm 1 details the training algorithm for DPGANs, which is effectively an instantiation of DPSGD. Note that only gradients for the discriminator $\mathcal{D}$ must be privatized (via clipping and noise), and not those for the generator $\mathcal{G}$. This is a consequence of closure under post-processing – the generator only interacts with the sensitive dataset indirectly via discriminator parameters, and therefore does not need further privatization.

## 4 Frequent discriminator steps improves private GANs

In this section, we discuss our main finding: $n_{\mathcal{D}}$, the number of discriminator steps taken between each generator step (see Algorithm 1) plays a significant role in the success of DPGAN training.

---

**Algorithm 1** TrainDPGAN$(D; \phi_0, \theta_0, \texttt{OptD}, \texttt{OptG}, n_{\mathcal{D}}, T, B, C, \sigma, \delta)$

---

1: **Input:** Labelled dataset $D = \{(x_j, y_j)\}_{j=1}^n$. Discriminator $\mathcal{D}$ and generator $\mathcal{G}$ initializations $\phi_0$ and $\theta_0$. Optimizers $\texttt{OptD}, \texttt{OptG}$. Hyperparameters: $n_{\mathcal{D}}$ ($\mathcal{D}$ steps per $\mathcal{G}$ step), $T$ (total number of $\mathcal{D}$ steps), $B$ (expected batch size), $C$ (clipping norm), and $\sigma$ (noise level). Privacy parameter $\delta$.
2: $q \leftarrow B/|D|$ and $t, k \leftarrow 0$ ▷ Calculate sampling rate $q$, initialize counters.
3: **while** $t < T$ **do** ▷ Update $\mathcal{D}$ with DPSGD.
4:     $S_t \sim \text{PoissonSample}(D, q)$ ▷ Sample a real batch $S_t$ by including each $(x, y) \in D$ w.p. $q$.
5:     $\tilde{S}_t \sim \mathcal{G}(\cdot; \theta_k)^B$ ▷ Sample fake batch $\tilde{S}_t$.
6:     $g_{\phi_t} \leftarrow \sum_{(x,y) \in S_t} \text{clip}\left(\nabla_{\phi_t}(-\log(\mathcal{D}(x, y; \phi_t))); C\right)$
          $+ \sum_{(\tilde{x},\tilde{y}) \in \tilde{S}_t} \text{clip}\left(\nabla_{\phi_t}(-\log(1 - \mathcal{D}(\tilde{x}, \tilde{y}; \phi_t))); C\right)$ ▷ Clip per-example gradients.
7:     $\widehat{g}_{\phi_t} \leftarrow \frac{1}{2B}(g_{\phi_t} + z_t)$, where $z_t \sim \mathcal{N}(0, C^2\sigma^2 I)$ ▷ Add Gaussian noise.
8:     $\phi_{t+1} \leftarrow \texttt{OptD}(\phi_t, \widehat{g}_{\theta_t})$
9:     $t \leftarrow t + 1$
10:     **if** $n_{\mathcal{D}}$ divides $t$ **then** ▷ Perform $\mathcal{G}$ update every $n_{\mathcal{D}}$ steps.
11:         $\tilde{S}_t' \sim \mathcal{G}(\cdot; \theta_k)^B$
12:         $g_{\theta_k} \leftarrow \frac{1}{B} \sum_{(\tilde{x},\tilde{y}) \in \tilde{S}_t'} \nabla_{\theta_k}(-\log(\mathcal{D}(\tilde{x}, \tilde{y}; \phi_t)))$
13:         $\theta_{k+1} \leftarrow \texttt{OptG}(\theta_k, g_{\theta_k})$
14:         $k \leftarrow k + 1$
15:     **end if**
16: **end while**
17: $\varepsilon \leftarrow \text{PrivacyAccountant}(T, \sigma, q, \delta)$ ▷ Compute privacy budget spent.
18: **Output:** Final $\mathcal{G}$ parameters $\theta_k$ and $(\varepsilon, \delta)$-DP guarantee.

---

Fixing a setting of DPSGD hyperparameters, there is an optimal range of values for $n_{\mathcal{D}}$ that maximizes generation quality, in terms of both visual quality and utility for downstream classifier training. This value can be quite large ($n_{\mathcal{D}} \approx 100$ in some cases).

### 4.1 Experimental details

**Setup.** We focus on labelled generation of MNIST (LeCun et al., 1998) and FashionMNIST (Xiao et al., 2017), both of which are comprised of 60K $28 \times 28$ grayscale images divided into 10 classes. To build a strong baseline, we begin from an open source PyTorch (Paszke et al., 2019) implementation[3] of DCGAN (Radford et al., 2016) that performs well non-privately, and copy their training recipe. We then adapt their architecture to our purposes: removing BatchNorm layers (which are not compatible with DPSGD) and adding label embedding layers to enable labelled generation. Training this configuration non-privately yields labelled generation that achieves FID scores of $3.4 \pm 0.1$ on MNIST and $16.5 \pm 1.7$ on FashionMNIST. $\mathcal{D}$ and $\mathcal{G}$ have 1.72M and 2.27M trainable parameters respectively. For further details, please see Appendix B.1.

**Privacy implementation.** To privatize training, we use Opacus (Yousefpour et al., 2021) which implements per-example gradient computation. As discussed before, we use the Rényi differential privacy (RDP) accounting of Mironov et al. (2019) (except in a few noted instances, where we instead use the tighter Gopi et al. (2021) accounting). For our baseline setting, we use the following DPSGD hyperparameters: we keep the non-private (expected) batch size $B = 128$, and use a noise level $\sigma = 1$ and clipping norm $C = 1$. Under these settings, we have the budget for $T = 450K$ discriminator steps when targeting $(10, 10^{-5})$-DP.

**Evaluation.** We evaluate our generative models by examining the *visual quality* and *utility for downstream tasks* of generated images. Following prior work, we measure visual quality by computing the Fréchet Inception Distance (FID) (Heusel et al., 2017) between 60K generated images and entire test set.[4] To measure downstream task utility, we again follow prior work, and train a CNN classifier on 60K generated image-label pairs and report its accuracy on the real test set.

---

[3]Courtesy of Hyeonwoo Kang (https://github.com/znxlwm). Code available at this link.
[4]We use an open source PyTorch implementation to compute FID: https://github.com/mseitzer/pytorch-fid.

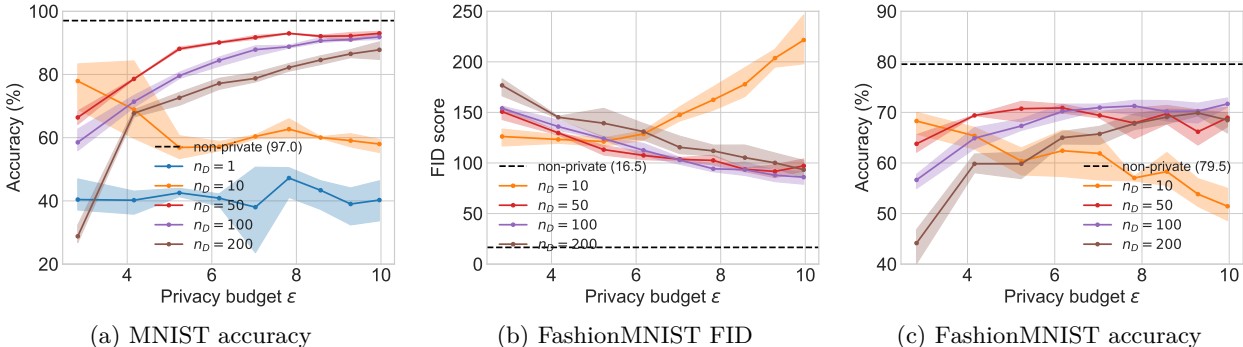

(a) MNIST accuracy      (b) FashionMNIST FID      (c) FashionMNIST accuracy

Figure 2: DPGAN results over training runs using different discriminator update frequencies $n_\mathcal{D}$, targeting $(10, 10^{-5})$-DP. Each plotted line indicates the mean, min, and max utility of 3 training runs with different seeds, as the privacy budget is expended. **(a)** We plot the test set accuracy of a CNN trained on generated data only. Accuracy mirrors the FID scores from Figure 1a. Going from $n_\mathcal{D} = 1$ to $n_\mathcal{D} = 50$ improves accuracy from $40.3 \pm 6.3\% \rightarrow 93.0 \pm 0.6\%$. Further $n_\mathcal{D}$ increases hurt accuracy. **(b) and (c)** We obtain similar results for FashionMNIST. Note that the optimal $n_\mathcal{D}$ is higher (around $n_\mathcal{D} \approx 100$). At $n_\mathcal{D} = 100$, we obtain an FID of $85.9 \pm 6.4$ and accuracy of $71.7 \pm 1.0\%$.

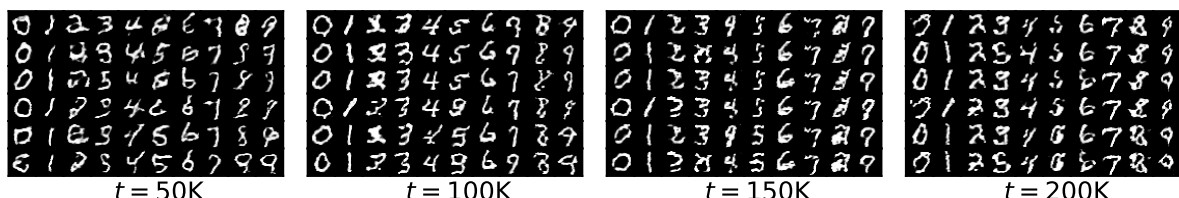

$t = 50K$      $t = 100K$      $t = 150K$      $t = 200K$

Figure 3: Evolution of samples drawn during training with $n_\mathcal{D} = 10$, when targeting $(10, 10^{-5})$-DP. This setting reports its best FID and downstream accuracy at $t = 50K$ iterations ($\varepsilon \approx 2.85$). As training progresses beyond this point, we observe mode collapse for several classes (e.g., the 6's and 7's, particularly at $t = 150K$), co-occuring with the deterioration in evaluation metrics (these samples correspond to the first 4 data points in the $n_\mathcal{D} = 10$ line in Figures 1a and 2a).

## 4.2 Results

**More frequent discriminator steps improves generation.** We plot in Figures 1a and 2 the evolution of FID and accuracy during DPGAN training for both MNIST and FashionMNIST, under varying discriminator update frequencies $n_\mathcal{D}$. The effect of this parameter has outsized impact on the final results. For MNIST, $n_\mathcal{D} = 50$ yields the best results; on FashionMNIST, $n_\mathcal{D} = 100$ is the best.

We emphasize that increasing the *frequency* of discriminator steps, relative to generator steps, does not affect the privacy cost of Algorithm 1. For any setting of $n_\mathcal{D}$, we perform the same number of noisy gradient queries on real data – what changes is the total number of generator steps taken over the course of training, which is reduced by a factor of $n_\mathcal{D}$.

**Private GANs are on a path to mode collapse.** For our MNIST results, we observe that at low discriminator update frequencies ($n_\mathcal{D} = 10$), the best FID and accuracy scores occur early in training, *well before the privacy budget we are targeting is exhausted.*[5] Examining Figures 1a and 2a at 50K discriminator steps (the leftmost points on the charts; $\varepsilon \approx 2.85$), the $n_\mathcal{D} = 10$ runs (in orange) have better FID and accuracy than both: (a) later checkpoints of the $n_\mathcal{D} = 10$ runs, after training longer and spending *more* privacy budget; and (b) other settings of $n_\mathcal{D}$ at that stage of training.

---

[5]This observation has been reported in Neunhoeffer et al. (2021), serving as motivation for their remedy of taking a mixture of intermediate models encountered in training. We are not aware of any mentions of this aspect of DPGAN training in papers reporting DPGAN baselines for labelled image synthesis.

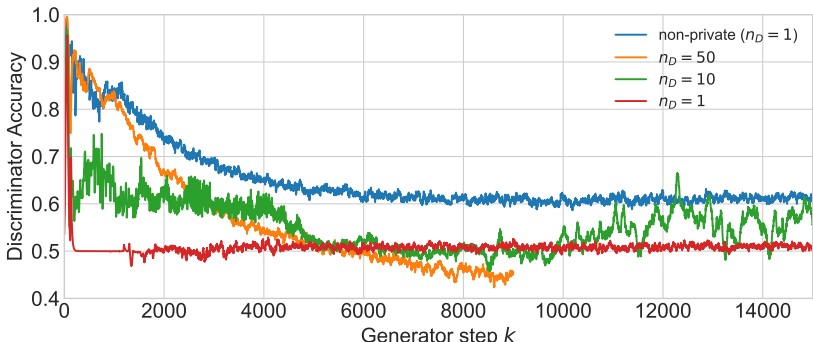

Figure 4: Exponential moving average ($\beta = 0.95$) of GAN discriminator accuracy on mini-batches, immediately before each generator step. While non-privately the discriminator maintains a 60% accuracy, the private discriminator with $n_\mathcal{D} = 1$ is effectively a random guess. Increasing the number of discriminator steps recovers the discriminator's advantage early on, leading to generator improvement. As the generator improves, the discriminator's task is made more difficult, driving down accuracy.

We attribute generator deterioration with more training to *mode collapse*: a known failure mode of GANs where the generator resorts to producing a small set of examples rather than representing the full variation present in the underlying data distribution. In Figure 3, we plot the evolution of generated images for an $n_\mathcal{D} = 10$ run over the course of training and observe qualitative evidence of mode collapse: at 50K steps, all generated images are varied, whereas at 150K steps, many of the columns (in particular the 6's and 7's) are slight variations of the same image. In contrast, successfully trained GANs do not exhibit this behaviour (see the $n_\mathcal{D} = 50$ images in Figure 1b). Mode collapse co-occurs with the deterioration in FID and accuracy observed in the first 4 data points of the $n_\mathcal{D} = 10$ runs (in orange) in Figures 1a and 2a.

**An optimal discriminator update frequency.** These results suggest that *fixing other DPSGD hyperparameters, there is an optimal setting for the discriminator step frequency $n_\mathcal{D}$* that strikes a balance between: (a) being too low, causing the generation quality to peak early in training and then undergo mode collapse; resulting in all subsequent training to consume additional privacy budget *without improving the model*; and (b) being too high, preventing the generator from taking enough steps to converge before the privacy budget is exhausted (an example of this is the $n_\mathcal{D} = 200$ run in Figure 2a). Striking this balance results in the most effective utilization of privacy budget towards improving the generator.

## 5    Why does taking more steps help?

In this section, we present empirical findings towards understanding why more frequent discriminator steps improves DPGAN training. We propose an explanation that is conistent with our findings.

**How does DP affect GAN training?** Figure 4 compares the accuracy of the GAN discriminator on held-out real and fake examples immediately before each generator step, between private and non-private training with different settings of $n_\mathcal{D}$. We observe that non-privately at $n_\mathcal{D} = 1$, discriminator accuracy stabilizes at around 60%. Naively introducing DP ($n_\mathcal{D} = 1$) leads to a qualitative difference: DP causes discriminator accuracy to drop to 50% (i.e., comparable accuracy to randomly guessing) immediately at the start of training, to never recover.[6]

For other settings of $n_\mathcal{D}$, we make following observations: (1) larger $n_\mathcal{D}$ corresponds to higher discriminator accuracy in early training; (2) in a training run, discriminator accuracy decreases throughout as the generator improves; (3) after discriminator accuracy falls below a certain threshold, the generator degrades or sees

---

[6]Our plot only shows the first 15K generator steps, but we remark that this persists until the end of training (450K steps).

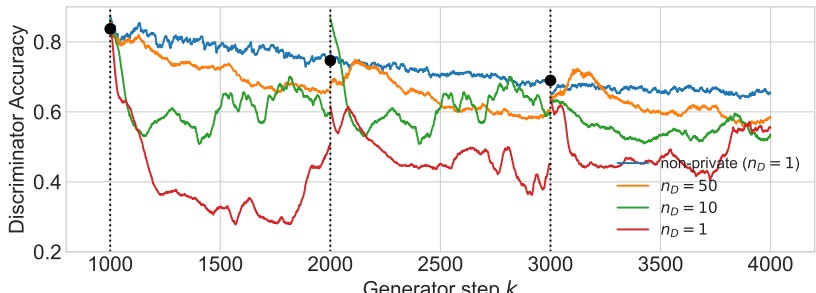

(a) Exponential moving average ($\beta = 0.95$) of discriminator accuracy on mini-batches, after checkpoint restarts

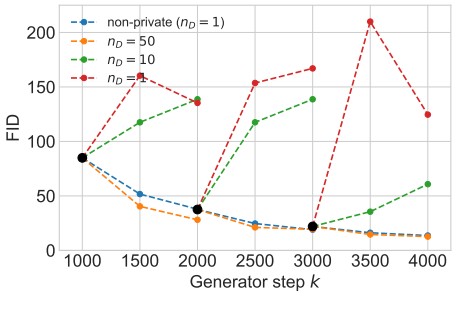

(b) FID after checkpoint restarts

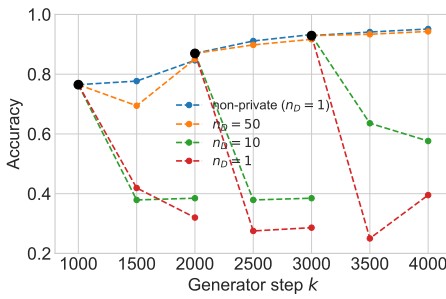

(c) Accuracy after checkpoint restarts

Figure 5: We restart training under various privacy and $n_\mathcal{D}$ settings at 3 checkpoints taken at 1K, 2K, and 3K generator steps into non-private training. We plot the progression of discriminator accuracy, FID, and downstream classification accuracy. The black dots correspond to the initial values of a checkpoint. We observe that low $n_\mathcal{D}$ settings do not achieve comparable discriminator accuracy to non-private training (a), and results in degradation of utility ((b) and (c)). Discriminator accuracy for $n_\mathcal{D} = 50$ tracks non-private training, and we observe utility improvement throughout training, as in the non-private setting.

limited improvement.[7] Based on these observations, we propose the following explanation for why more frequent discriminator steps help:

- Generator improvement occurs when the discriminator is effective at distinguishing between real and fake data.

- The *asymmetric noise addition* introduced by DP to the discriminator makes such a task difficult, resulting in limited generator improvement.

- Allowing the discriminator to train longer on a fixed generator improves its accuracy, recovering the non-private case where the generator and discriminator are balanced.

**Checkpoint restarting experiment.** We perform a checkpoint restarting experiment to examine this explanation in a more controlled setting. We train a non-private GAN for 3K generator steps, and save checkpoints of $\mathcal{D}$ and $\mathcal{G}$ (and their respective optimizers) at 1K, 2K, and 3K generator steps. We restart training from each of these checkpoints for 1K generator steps under different $n_\mathcal{D}$ and privacy settings. We plot the progression of discriminator accuracy, FID, and downstream classification accuracy. Results are pictured in Figure 5. Broadly, our results corroborate the observations that discriminator accuracy improves with larger $n_\mathcal{D}$ and decreases with better generators, and that generator improvement occurs when the discriminator has sufficiently high accuracy.

---

[7]For $n_\mathcal{D} = 10$, accuracy falls below 50% after 5K $\mathcal{G}$ steps (= 50K $\mathcal{D}$ steps), which corresponds to the first point in the $n_\mathcal{D} = 10$ line in Figures 1a and 2a. For $n_\mathcal{D} = 50$, accuracy falls below 50% after 5K $\mathcal{G}$ steps (= 250K $\mathcal{D}$ steps), which corresponds to the 5th point in the $n_\mathcal{D} = 50$ line in Figures 1a and 2a.

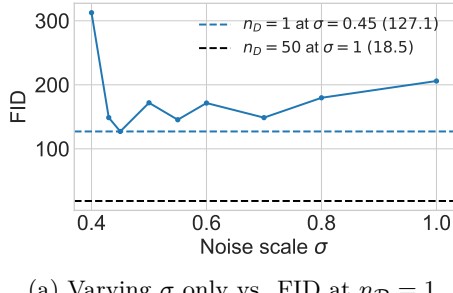
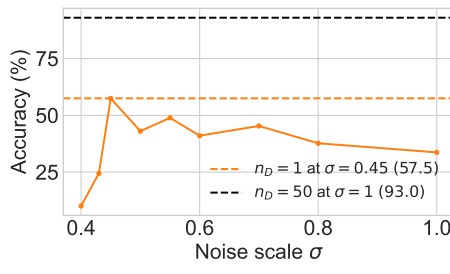

(a) Varying $\sigma$ only vs. FID at $n_{\mathcal{D}} = 1$  (b) Varying $\sigma$ only vs. accuracy at $n_{\mathcal{D}} = 1$

Figure 6: On MNIST, we fix $n_{\mathcal{D}} = 1$ and report results for various settings of the DPSGD noise level $\sigma$, where the number of iterations $T$ is chosen for each $\sigma$ to target $(10, 10^{-5})$-DP. The gap between the dashed lines represent the advancement of the utility frontier by incorporating the choice of $n_{\mathcal{D}}$ into our design space.

**Does reducing noise accomplish the same thing?**    In light of the above explanation, we ask if reducing the noise level $\sigma$ can offer the same improvement as taking more steps, as reducing $\sigma$ should also improve discriminator accuracy before a generator step. To test this: starting from our setting in Section 4, fixing $n_{\mathcal{D}} = 1$, and targeting MNIST at $\varepsilon = 10$, we search over a grid of noise levels $\sigma$ (the lowest of which, $\sigma = 0.4$, admits a budget of only $T = 360$ discriminator steps). Results are pictured in Figure 6. We obtain a best FID of 127.1 and best accuracy of 57.5% at noise level $\sigma = 0.45$. Hence we can conclude that in this experimental setting, incorporating discriminator update frequency in our design space allows for more effective use of privacy budget for improving generation quality.

**Does taking more discriminator steps always help?**    As we discuss in more detail in Section 6.1, when we are able to find other means to improve the discriminator beyond taking more steps, tuning discriminator update frequency may not yield improvements. To illustrate with an extreme case, consider eliminating the privacy constraint. In non-private GAN training, taking more steps is known to be unnecessary. We corroborate this result: we run our non-private baseline from Section 4 with the same number of generator steps, but opt to take 10 discriminator steps between each generator step instead of 1. FID worsens from $3.4 \pm 0.1 \rightarrow 8.3$, and accuracy worsens from $97.0 \pm 0.1\% \rightarrow 91.3\%$.

## 6 Better generators via better discriminators

Our proposed explanation in Section 5 provides a concrete suggestion for improving GAN training: effectively use our privacy budget to maximize the number of generator steps taken when the discriminator has sufficiently high accuracy. We experiment with modifications to the private GAN training recipe towards these ends, which translate to improved generation.

### 6.1 Larger batch sizes

Several recent works have demonstrated that for classification tasks, DPSGD achieves higher accuracy with larger batch sizes, after tuning the noise level $\sigma$ accordingly (Tramèr & Boneh, 2021; Anil et al., 2022; De et al., 2022). On simpler, less diverse datasets (such as MNIST, CIFAR-10, and FFHQ), GAN training is typically conducted with small batch sizes (for example, DCGAN uses $B = 128$ (Radford et al., 2016), which we adopt; StyleGAN(2|3) uses $B = 32/64$ (Karras et al., 2019; 2020; 2021)).[8] Therefore it is interesting to see if large batch sizes help in our setting. We corroborate that on MNIST, larger batch sizes do not significantly improve our non-private baseline from Section 4: when we go up to $B = 2048$ from $B = 128$, FID goes from $3.4 \pm 0.1 \rightarrow 3.2$ and accuracy goes from $97.0 \pm 0.1\% \rightarrow 97.5\%$.

---

[8]However on more complex, diverse datasets (such as ImageNet), it has been found that larger batch sizes help: this is the conclusion from BigGAN (Brock et al., 2019). Recent work scaling up StyleGAN to diverse datasets – StyleGAN-XL (Sauer et al., 2022) and GigaGAN (Kang et al., 2023) – corroborate this result, seeing improvements from scaling up batch sizes to 2048 and 1024 respectively.

| Privacy $\varepsilon$ | Method | Reported in | MNIST | | FashionMNIST | |
|---|---|---|---|---|---|---|
| | | | FID | Acc.(%) | FID | Acc.(%) |
| $\varepsilon = \infty$ | Real data | This work | 1.0 | 99.2 | 1.5 | 92.5 |
| | GAN | | $3.4 \pm 0.1$ | $97.0 \pm 0.1$ | $16.5 \pm 1.7$ | $79.5 \pm 0.8$ |
| $\varepsilon = 10$ | DP-MERF | Cao et al. (2021) | 116.3 | 82.1 | 132.6 | 75.5 |
| | DP-Sinkhorn | Cao et al. (2021) | 48.4 | 83.2 | 128.3 | 75.1 |
| | PSG[9] | Chen et al. (2022) | - | 95.6 | - | 77.7 |
| | DPDM | Dockhorn et al. (2022) | 5.01 | 97.3 | 18.6 | 84.9 |
| | DPGAN[10] | Chen et al. (2020) | 179.16 | 63 | 243.80 | 50 |
| | | Long et al. (2021) | 304.86 | 80.11 | 433.38 | 60.98 |
| | GS-WGAN | Chen et al. (2020) | 61.34 | 80 | 131.34 | 65 |
| | PATE-GAN | Long et al. (2021) | 253.55 | 66.67 | 229.25 | 62.18 |
| | G-PATE | Long et al. (2021) | 150.62 | 80.92 | 171.90 | 69.34 |
| | DataLens | Wang et al. (2021) | 173.50 | 80.66 | 167.68 | 70.61 |
| $\varepsilon = 9.32^*$ | Our DPGAN | | $18.5 \pm 0.9$ | $93.0 \pm 0.6$ | $85.9 \pm 6.4$ | $71.7 \pm 1.0$ |
| | + large batches | This work | $13.2 \pm 1.1$ | $94.0 \pm 0.6$ | $70.9 \pm 6.3$ | $73.0 \pm 1.1$ |
| | + adaptive $n_{\mathcal{D}}$ | | $12.8 \pm 0.3$ | $95.1 \pm 0.1$ | $62.3 \pm 8.7$ | $74.7 \pm 0.4$ |
| $\varepsilon = 1$ | DP-MERF[11] | Vinaroz et al. (2022) | - | 80.7 | - | 73.9 |
| | DP-HP | Vinaroz et al. (2022) | - | 81.5 | - | 72.3 |
| | PSG | Chen et al. (2022) | - | 80.9 | - | 70.2 |
| | DPDM | Dockhorn et al. (2022) | 23.4 | 95.3 | 37.8 | 79.4 |
| | DPGAN | Long et al. (2021) | 470.20 | 40.36 | 472.03 | 10.53 |
| | GS-WGAN | Long et al. (2021) | 489.75 | 14.32 | 587.31 | 16.61 |
| | PATE-GAN | Long et al. (2021) | 231.54 | 41.68 | 253.19 | 42.22 |
| | G-PATE | Long et al. (2021) | 153.38 | 58.80 | 214.78 | 58.12 |
| | DataLens | Wang et al. (2021) | 186.06 | 71.23 | 194.98 | 64.78 |
| $\varepsilon = 0.912^*$ | Our DPGAN | | $111.1 \pm 17.9$ | $76.9 \pm 0.6$ | $155.3 \pm 7.1$ | $64.9 \pm 0.8$ |
| | + large batches | This work | $106.2 \pm 64.0$ | $67.5 \pm 7.8$ | $158.9 \pm 6.0$ | $67.2 \pm 1.0$ |
| | + adaptive $n_{\mathcal{D}}$ | | $52.6 \pm 3.2$ | $81.3 \pm 0.8$ | $126.4 \pm 4.1$ | $69.1 \pm 0.1$ |

Table 2: We gather previously reported results in the literature on the performance of various methods for labelled generation of MNIST and FashionMNIST, compared with our results. For our results, we run 3 seeds and report mean $\pm$ std. Note that *Reported In* refers to the source of the numerical result, not the originator of the approach. For downstream accuracy, we report the best accuracy among classifiers they use, and compare against our CNN classifier accuracy. (*) For our results, we target $\varepsilon = 10/\varepsilon = 1$ with Opacus accounting and additionally report $\varepsilon$ using the improved privacy accounting of Gopi et al. (2021).

**Results.** We scale up batch sizes, considering $B \in \{128, 512, 2048\}$, and search for the optimal noise level $\sigma$ and $n_{\mathcal{D}}$ (details in Appendix B.2). We target both $\varepsilon = 1$ and $\varepsilon = 10$. We report the best results from our hyperparameter search in in Table 2. We find that larger batch sizes leads to improvements: for $\varepsilon = 1$ and $\varepsilon = 10$, the best results are achieved at $B = 512$ and $B = 2048$ respectively. We also note that for large batch sizes, the optimal number of generator steps can be quite small. For $B = 2048$, $\sigma = 4.0$, targeting MNIST at $\varepsilon = 10$, $n_{\mathcal{D}} = 5$ is the optimal discriminator update frequency, and improves over our best $B = 128$ setting employing $n_{\mathcal{D}} = 50$. For full results, see Appendix D.3.

## 6.2 Adaptive discriminator step frequency

Our observations from Section 4 and 5 motivate us to consider *adaptive* discriminator step frequencies. As pictured in Figures 4 and 5a, discriminator accuracy drops during training as the generator improves. In

---

[9]Since PSG produces a coreset of only 200 examples (20 per class), the covariance of its InceptionNet-extracted features is singular, and therefore it is not possible to compute an FID score.

[10]We group per-class unconditional GANs together with conditional GANs under the DPGAN umbrella.

[11]Results from Vinaroz et al. (2022) are presented graphically in the paper. Exact numbers can be found in their code.

| Privacy | Method | Reported In | FID | Acc.(%) |
|---|---|---|---|---|
| $\varepsilon = \infty$ | Real data
GAN | This work | 1.1
$30.0 \pm 1.6$ | 96.6
$92.0 \pm 0.4$ |
| $\varepsilon = 10$ | DP-MERF
DP-Sinkhorn
DPDM | Cao et al. (2021)
Cao et al. (2021)
Dockhorn et al. (2022) | 274.0
189.5
21.1 | 65
76.3
- |
| | DPGAN
GS-WGAN
PATE-GAN
G-PATE | Long et al. (2021)
Long et al. (2021)
Long et al. (2021)
Long et al. (2021) | -
-
-
- | 54.09
63.26
58.70
70.72 |
| $\varepsilon = 9.39^*$ | Our DPGAN | This work | $170.8 \pm 20.3$ | $82.4 \pm 4.4$ |
| $\varepsilon = 10$ | DPGAN
GS-WGAN
PATE-GAN
G-PATE
DataLens | Long et al. (2021)
Long et al. (2021)
Long et al. (2021)
Long et al. (2021)
Wang et al. (2021) | 485.41
432.58
424.60
305.92
320.84 | 52.11
61.36
65.35
68.97
72.87 |

Table 3: **Top section of the table:** Comparison to state-of-the-art results on $32 \times 32$ CelebA-Gender, targeting $(\varepsilon, 10^{-6})$-DP (except for the results reported in Long et al. (2021) which target a weaker $(\varepsilon, 10^{-5})$-DP). We run 3 seeds and report the mean $\pm$ std. **(*)** For our results, we target $\varepsilon = 10$ with Opacus accounting and additionally report $\varepsilon$ using the improved privacy accounting of Gopi et al. (2021). DPDM reports a much better FID score than our DPGAN (which itself, is an improvement over previous results). Our DPGAN achieves the best reported accuracy score. **Bottom section of the table:** Results for GAN-based approaches reported in Long et al. (2021) and Wang et al. (2021), which are not directly comparable because they target $(10, 10^{-5})$-DP and use $64 \times 64$ CelebA-Gender.

this scenario, we want to take more steps to improve the discriminator, in order to further improve the generator. However, using a large discriminator update frequency right from the beginning of training is wasteful – as evidenced by the fact that low $n_{\mathcal{D}}$ achieves the best FID and accuracy early in training. Hence we propose to start at a low discriminator update frequency ($n_{\mathcal{D}} = 1$), and ramp up when our discriminator is performing poorly. Accuracy on real data must be released with DP. While this is feasible, it introduces the additional problem of having to find the right split of privacy budget for the best performance. We observe that discriminator accuracy is related to discriminator accuracy on fake samples only (which are free to evaluate on, by post-processing). Hence we use it as a proxy to assess discriminator performance.

We propose an *adaptive step frequency*, parameterized by $\beta$ and $d$. $\beta$ is the decay parameter used to compute the exponential moving average (EMA) of discriminator accuracy on fake batches before each generator update. $d$ is the accuracy floor that upon reaching, we move to the next update frequency $n_{\mathcal{D}} \in \{1, 2, 5, 10, 20, 50, 100, 200, 500, 1000, ...\}$. Additionally, we promise a grace period of $2/(1-\beta)$ generator steps before moving on to the next update frequency – motivated by the fact that $\beta$-EMA's value is primarily determined by its last $2/(1 - \beta)$ observations.

We use $\beta = 0.99$ in all settings, and try $d = 0.6$ and $d = 0.7$. The additional benefit of the adaptive step frequency is that it means we do not have to search for the optimal update frequency. Although the adaptive step frequency introduces the extra hyperparameter of the threshold $d$, we found that these two settings ($d = 0.6$ and $d = 0.7$) were sufficient to improve over results of a much more extensive hyperparameter search over $n_{\mathcal{D}}$ (whose optimal value varied significantly based on the noise level $\sigma$ and expected batch size $B$).

### 6.3 Comparison with previous results in the literature

### 6.3.1 MNIST and FashionMNIST

Table 2 summarizes our best experimental settings for MNIST and FashionMNIST, and situates them in the context of previously reported results for the task. We also present a visual comparison in Figure 7. We provide some examples of generated images in Figures 9 and 10 for $\varepsilon = 10$, and Figures 11 and 12 for $\varepsilon = 1$.

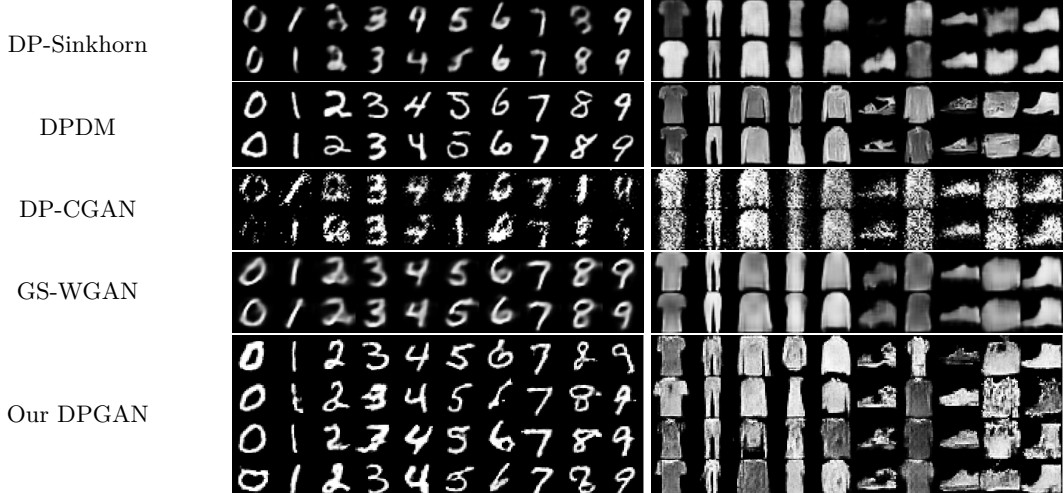

DP-Sinkhorn

DPDM

DP-CGAN

GS-WGAN

Our DPGAN

Figure 7: MNIST and FashionMNIST results at $(10, 10^{-5})$-DP for different methods. Images of other methods are from Cao et al. (2021) and Dockhorn et al. (2022).

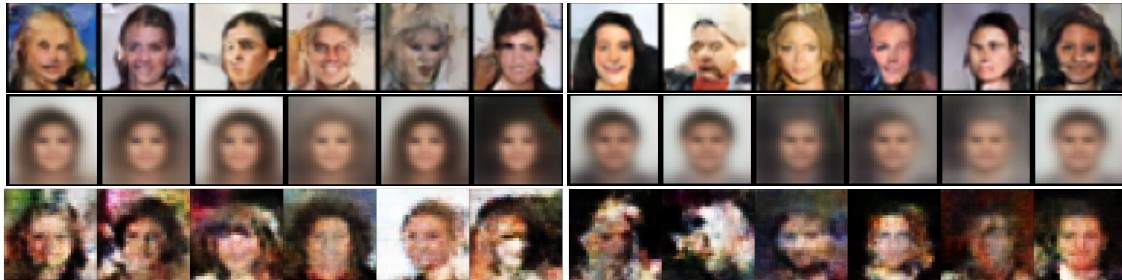

Figure 8: $32 \times 32$ CelebA-Gender at $(10, 10^{-6})$-DP. **From top to bottom:** DPDM (unconditional generation), DP-Sinkhorn, and our DPGAN. Images of other methods are from Cao et al. (2021) and Dockhorn et al. (2022).

**Plain DPSGD beats all alternative GAN privatization schemes.** Our baseline DPGAN from Section 4, with the appropriate choice of $n_{\mathcal{D}}$ (and without the modifications described in this section yet), outperforms all other GAN-based approaches proposed in the literature (GS-WGAN, PATE-GAN, G-PATE, and DataLens) *uniformly* across both metrics, both datasets, and both privacy levels.

**Large batch sizes and adaptive discriminator step frequency improve GAN training.** Broadly speaking, across both privacy levels and both datasets, we see an improvement from taking larger batch sizes, and then another with the adaptive step frequency.

**Comparison with state-of-the-art.** With the exception of DPDM, our best DPGANs are competitive with state-of-the-art approaches for DP synthetic data, especially in terms of FID scores.

### 6.3.2 CelebA-Gender

We also report results on generating $32 \times 32$ CelebA, conditioned on gender at $(10, 10^{-6})$-DP. For these experiments, we employed large batches ($B = 2048$) and adaptive discriminator step frequency with threshold $d = 0.6$. Full implementation details can be found in Appendix C. Results are summarized in Table 3 and visualized in Figure 8. For more example generations, see Figure 13.

# 7 Conclusion

We revisit differentially private GANs and show that, with appropriate tuning of the training procedure, they can perform dramatically better than previously thought. Some crucial modifications include increasing discriminator step frequency, increasing the batch size, and introducing adaptive discriminator step frequency. We explore the hypothesis that the previous deficiencies of DPGANs were due to poor classification accuracy of the discriminator. More broadly, our work supports the recurring finding that carefully-tuned DPSGD on conventional architectures can yield strong results for differentially private machine learning.

## Acknowledgements

AB is supported by an NSERC Discovery Grant, a David R. Cheriton Graduate Scholarship, and an Ontario Graduate Scholarship. GK is supported by an NSERC Discovery Grant, an unrestricted gift from Google, and a University of Waterloo startup grant. We would like to thank the TMLR anonymous reviewers and action editor for providing constructive feedback.

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

## A    Generated samples

We provide a few non-cherrypicked samples for MNIST and FashionMNIST at $\varepsilon = 10$ and $\varepsilon = 1$, as well as $32 \times 32$ CelebA-Gender at $\varepsilon = 10$.

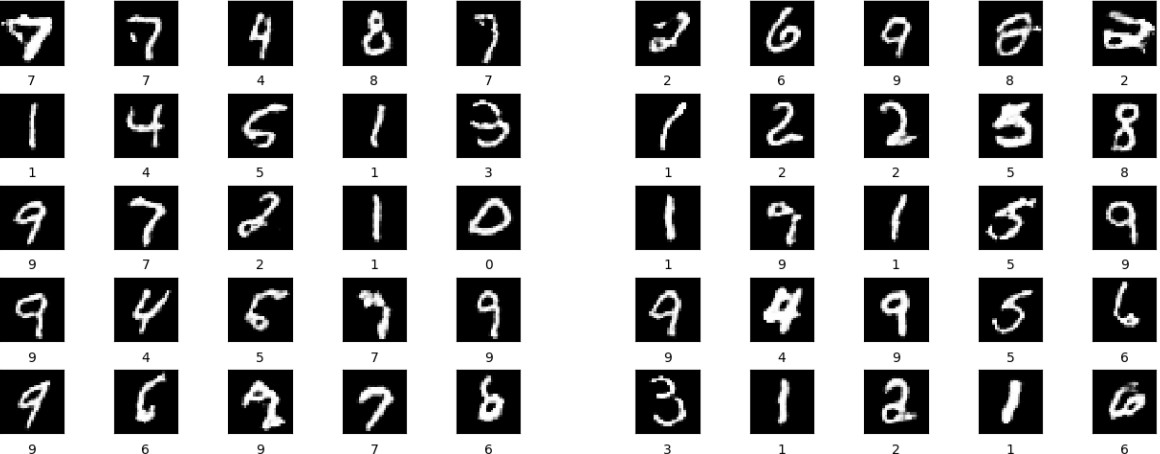

Figure 9: Some non-cherrypicked MNIST samples from our method, $\varepsilon = 10$.

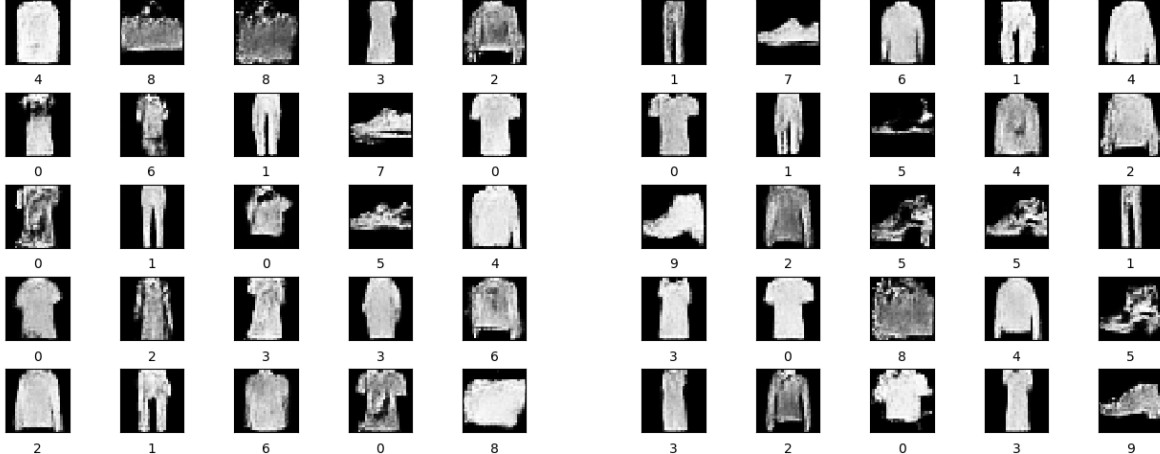

Figure 10: Some non-cherrypicked FashionMNIST samples from our method, $\varepsilon = 10$.

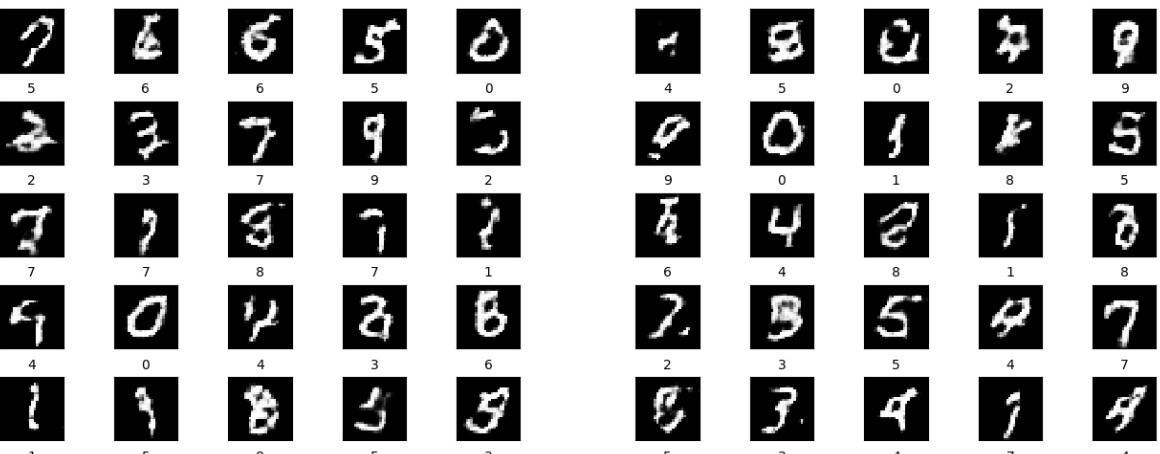

Figure 11: Some non-cherrypicked MNIST samples from our method, $\varepsilon = 1$.

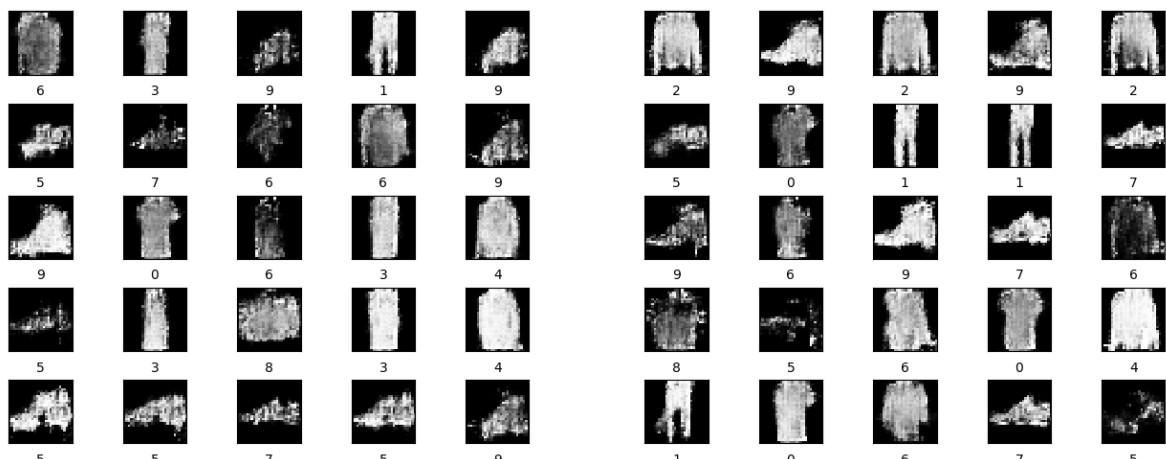

Figure 12: Some non-cherrypicked FashionMNIST samples from our method, $\varepsilon = 1$.

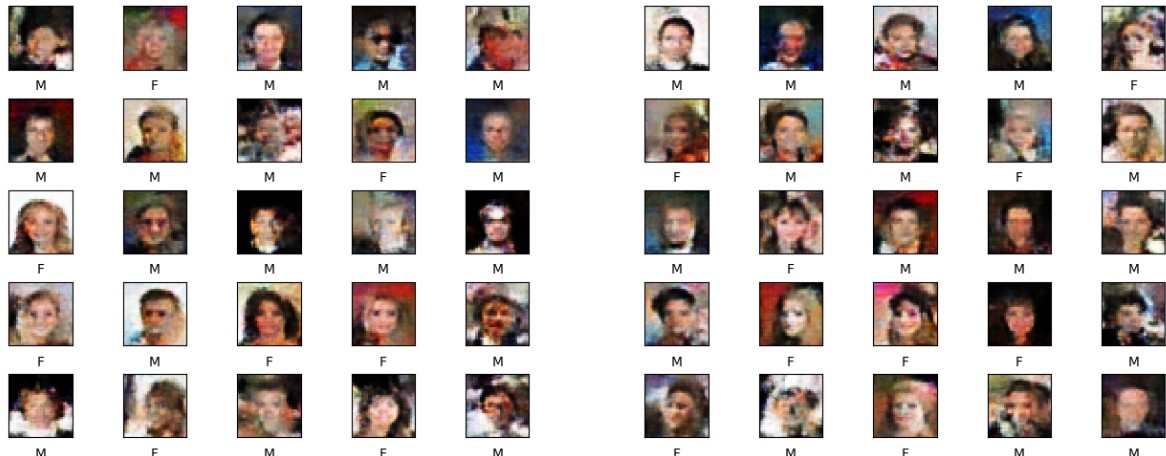

Figure 13: Some non-cherrypicked CelebA samples from our method, $\varepsilon = 10$.

# B MNIST and FashionMNIST implementation details

## B.1 Training recipe

For MNIST and FashionMNIST, we begin from an open source PyTorch implementation of DCGAN (Radford et al., 2016) (available at this link) that performs well non-privately, and copy their training recipe. This includes: batch size $B = 128$, the Adam optimizer (Kingma & Ba, 2015) with parameters ($\alpha = 0.0002, \beta_1 = 0.5, \beta_2 = 0.999$) for both $\mathcal{G}$ and $\mathcal{D}$, the non-saturating GAN loss (Goodfellow et al., 2014), and a 5-layer fully convolutional architecture with width parameter $d = 128$.

To adapt it to our purposes, we make three architectural modifications: in both $\mathcal{G}$ and $\mathcal{D}$ we (1) remove all BatchNorm layers (which are not compatible with DPSGD); (2) add label embedding layers to enable labelled generation; and (3) adjust convolutional/transpose convolutional stride lengths and kernel sizes as well as remove the last layer, in order to process $1 \times 28 \times 28$ images without having to resize. Finally, we remove their custom weight initialization, opting for PyTorch defaults.

Our baseline non-private GANs are trained for 45K steps. We train our non-private GANs with poisson sampling as well: for each step of discriminator training, we sample real examples by including each element of our dataset independently with probability $B/n$, where $n$ is the size of our dataset. We then add $B$ fake examples sampled from $\mathcal{G}$ to form our fake/real combined batch.

**Clipping fake sample gradients.** When training the discriminator privately with DPSGD, we draw $B$ fake examples and compute clipped per-example gradients on the entire combined batch of real and fake examples (see Algorithm 1). This is the approach taken in the prior work of Torkzadehmahani et al. (2019). We remark that this is purely a design choice – it is not necessary to clip the gradients of the fake samples, nor to process them together in the same batch. So long as we preserve the sensitivity of gradient queries *with respect to the real data*, the same amount of noise will suffice for privacy.

## B.2 Large batch size hyperparameter search

We scale up batch sizes, considering $B \in \{64, 128, 512, 2048\}$, and search for the optimal noise level $\sigma$ and $n_{\mathcal{D}}$. For $B = 128$ targeting $\varepsilon = 10$, we search over three noise levels $\Sigma_{128}^{10} = \{0.6, 1.0, 1.4\}$. We choose candidate noise levels for other batch sizes as follows: when considering a batch size $B = 128n$, we search over

$$\Sigma_{128n}^{10} := \{\sqrt{n} \cdot \sigma : \sigma \in \Sigma_{128}^{10}\}.$$

We also target the high privacy ($\varepsilon = 1$) regime. For $\varepsilon = 1$, we multiply all noise levels by 5,

$$\Sigma_B^1 = \{5\sigma : \sigma \in \Sigma_B^{10}\}.$$

For each setting of $(B, \sigma)$, we search over a grid of $n_{\mathcal{D}} \in \{1, 2, 5, 10, 20, 50, 100, 200, 500\}$. Due to compute limitations, we omit some values that we are confident will fail (e.g., trying $n_{\mathcal{D}} = 1$ when mode collapse occurs for $n_{\mathcal{D}} = 5$).

## C  CelebA implementation details

The CelebA dataset (Liu et al., 2015) consists of 202,599 178×218 RGB images of celebrity faces, each labelled with 40 binary attributes. The version of the dataset we work with, 32x32 CelebA-Gender (a benchmark reported in Cao et al. (2021)), is obtained by resizing to 32x32 and labelling with the gender attribute. The 202,599 images are partitioned into a training set of size 182,637 and a test set of size 19,962.

We use essentially the same model architectures we used for MNIST and FashionMNIST for CelebA: 4-layer fully convolutional networks with label embedding layers for both $\mathcal{D}$ and $\mathcal{G}$. We adjust convolutional/transpose convolutional stride lengths and kernels sizes to process $3 \times 32 \times 32$ images without having to resize. $\mathcal{D}$ and $\mathcal{G}$ for CelebA are slightly larger, having 2.64M and 3.16M trainable parameters respectively.

Drawing from the results of our MNIST and FashionMNIST experiments, we used a large batch size ($B = 2048$) and adaptive discriminator updates, with threshold $d = 0.6$. We experimented with a few settings for noise level $\sigma \in \{2, 3, 4\}$. Our best results were with the largest noise $\sigma = 4$ which gave us 385K discriminator steps when targeting $\varepsilon = 10$.

## D  Ablations

### D.1  Varying discriminator size

We train DPGANs on MNIST under the setting of Section 4: using noise level $\sigma = 1$, batch size $B = 128$, and targeting $\varepsilon = 10$ which yields 450K discriminator steps. By adjusting $d_{\mathcal{D}}$ (the # of filter banks in the first convolutional layer of the discriminator, which controls width throughout), we can obtain discriminators with roughly $0.25\times$, $0.5\times$, and $2\times$ the parameter count (Table 4). For these experiments, we vary discriminator size while keeping the generator size (2.27M parameters) fixed.

| $d_{\mathcal{D}}$ | $\mathcal{D}$ parameter count | Ratio |
|---|---|---|
| 64 | 0.44M | $0.26\times$ |
| 96 | 0.97M | $0.57\times$ |
| 128 | 1.72M | $1\times$ |
| 196 | 3.86M | $2.24\times$ |

Table 4: Number of trainable parameters in discriminator size variants.

**Results.**  In Figure 14 we plot the progression of FID and downstream classifier accuracy of generated MNIST samples during non-private training with discriminators of varying size. We observe that, non-privately, larger discriminators do better in terms of FID early on, and converge to slightly worse accuracies.

In Figures 15 and 16, we plot the progression of FID and accuracy (respectively) for DPGANs trained on MNIST (targeting $\varepsilon = 10$) at different discriminator update frequencies $n_{\mathcal{D}}$. In each plot, we compare the $d_{\mathcal{D}} = 128$ runs (in green), which correspond to results from Figures 1a and 2a, to the results of training with discriminators with 0.26–2.24× as many trainable parameters. These additional settings mostly track the $d_{\mathcal{D}} = 128$ runs. Larger discriminators appear to perform slightly better, especially in terms of accuracy early in training. Larger discriminators also use significantly more compute.

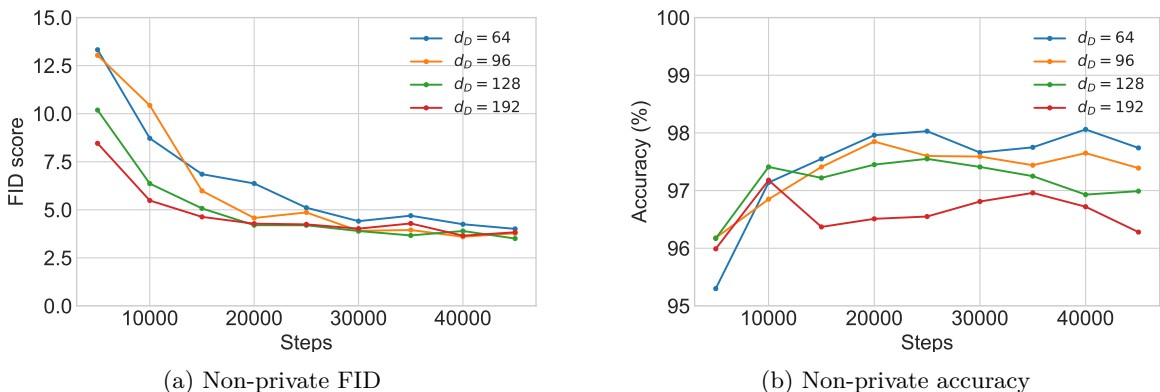

(a) Non-private FID           (b) Non-private accuracy

Figure 14: MNIST FID and downstream classifier accuracy for non-private GAN training with various discriminator sizes. The green ($d_{\mathcal{D}} = 128$) line corresponds to the 1.72M parameter discriminator used in previous experiments.

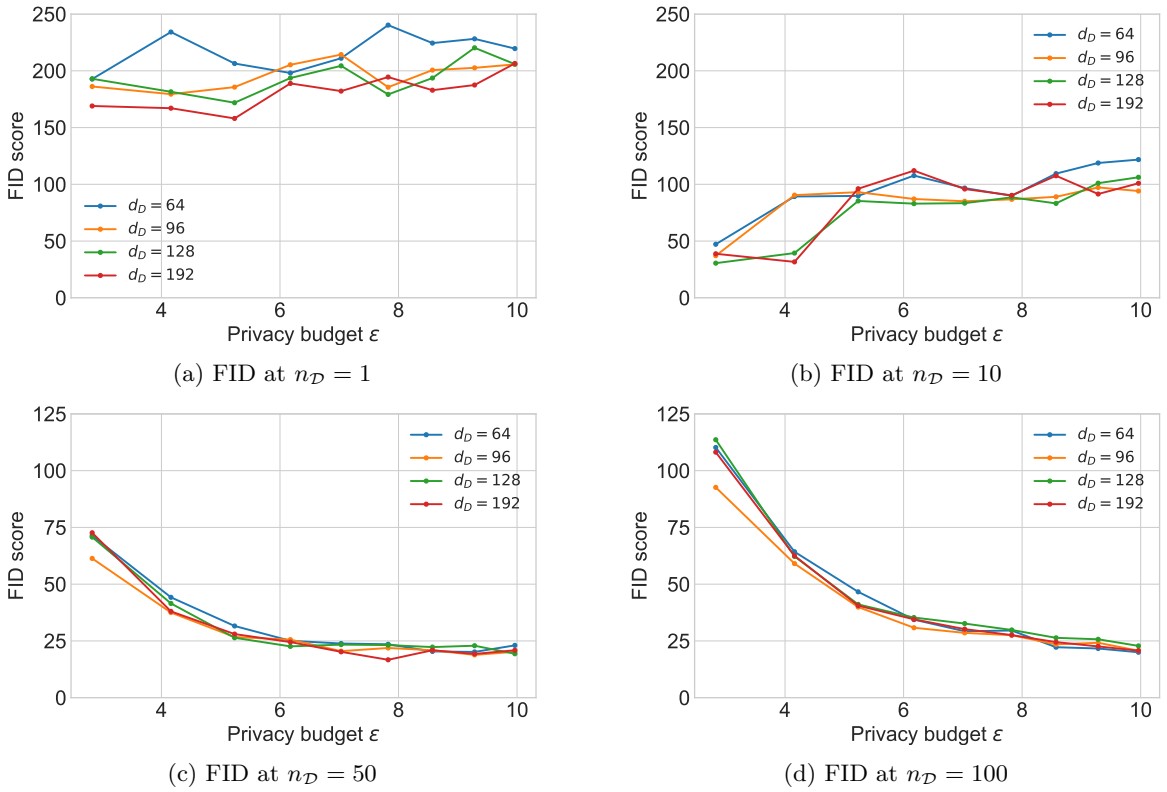

(a) FID at $n_{\mathcal{D}} = 1$           (b) FID at $n_{\mathcal{D}} = 10$

(c) FID at $n_{\mathcal{D}} = 50$           (d) FID at $n_{\mathcal{D}} = 100$

Figure 15: MNIST FID for DPGAN training (targeting $\varepsilon = 10$) at various discriminator update frequencies $n_{\mathcal{D}}$. In each plot, we present results from training with various discriminator sizes. The green ($d_{\mathcal{D}} = 128$) lines correspond to the results pictured in Figure 1a. Discriminators with 0.26–2.24× as many trainable parameters track the results of the original $d_{\mathcal{D}} = 128$ setting.

## D.2 Varying learning rate

We train DPGANs on MNIST under the setting of Section 4: using noise level $\sigma = 1$, batch size $B = 128$, and targeting $\varepsilon = 10$ which yields 450K discriminator steps. Here, we keep discriminator size $d_{\mathcal{D}} = 128$ fixed, and vary the learning rates of $\mathcal{G}$ and $\mathcal{D}$, while keeping the other Adam parameters $\beta_1$ and $\beta_2$ for both $\mathcal{G}$ and $\mathcal{D}$ fixed. Table 5 lists the learning rate settings we consider.

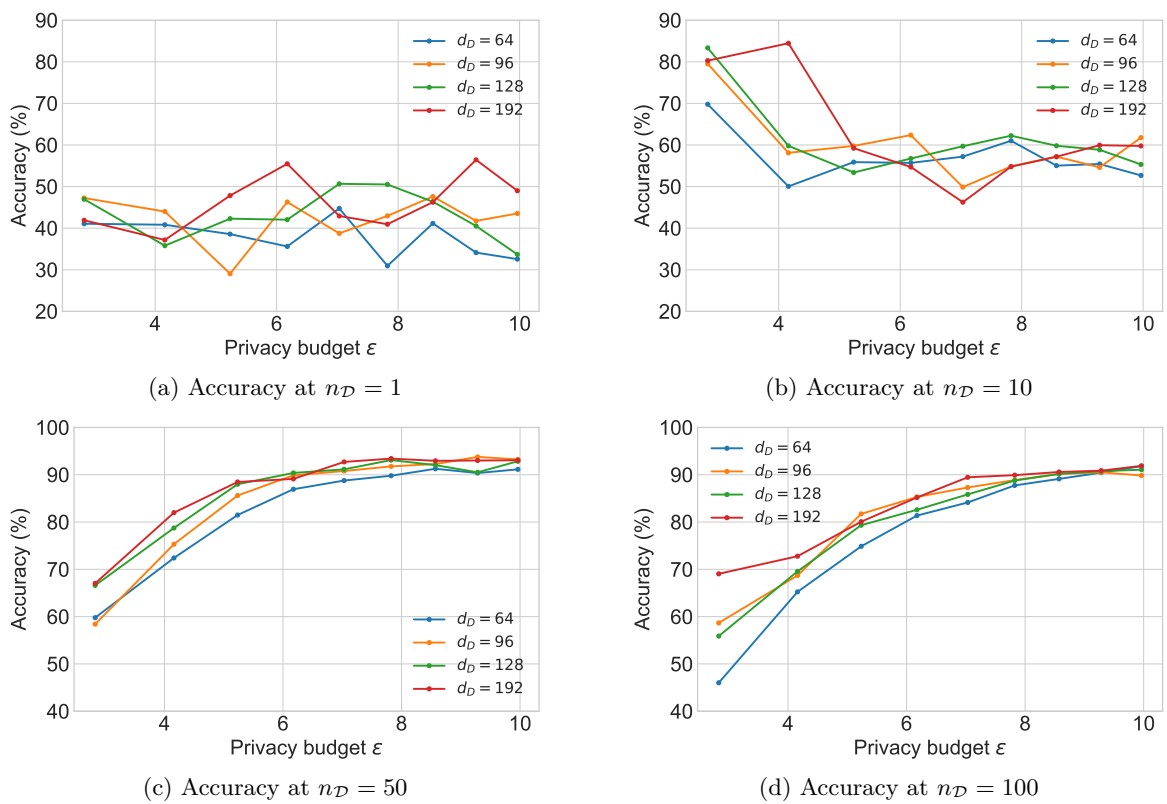

Figure 16: MNIST downstream classifier accuracy for DPGAN training (targeting $\varepsilon = 10$) at various discriminator update frequencies $n_{\mathcal{D}}$. In each plot, we present results from training with various discriminator sizes. The green ($d_{\mathcal{D}} = 128$) lines correspond to the results pictured in Figure 2a. Discriminators with 0.26–2.24$\times$ as many trainable parameters track the results of the original $d_{\mathcal{D}} = 128$ setting.

| Setting | $\mathcal{G}$ LR | $\mathcal{D}$ LR |
|---|---|---|
| Base | 0.0002 | 0.0002 |
| 5$\times$ LR | 0.001 | 0.001 |
| 0.2$\times$ LR | 0.00004 | 0.00004 |
| 5$\times$ $\mathcal{D}$ LR | 0.0002 | 0.001 |
| 0.2$\times$ $\mathcal{D}$ LR | 0.0002 | 0.00004 |

Table 5: Learning rate settings.

**Results.** In Figure 17 we plot the progression of FID and downstream classifer accuracy of generated MNIST samples during non-private training under various learning rate settings. We observe FID and accuracy degradation near the end of training for the 5$\times$ ($\mathcal{D}$) LR settings. The 0.2$\times$ LR setting converges much slower. This is remedied when we adjust *only the $\mathcal{D}$ LR* by 0.2$\times$ and leave $\mathcal{G}$ LR unchanged (comparing the green line to the purple line in Figure 17).

Figures 18 and 19 examine the case where we adjust both $\mathcal{G}$ and $\mathcal{D}$ learning rates by 5$\times$ and 0.2$\times$ respectively. Broadly, we see the same behaviour in Section 4: FID and downstream classification accuracy improve significantly as we take $n_{\mathcal{D}} >> 1$, up until the point where $n_{\mathcal{D}}$ is too high, limiting the generator from taking enough steps to converge. However, we note some differences: (1) the performance of the best settings for $n_{\mathcal{D}}$ are reduced across the board (most prominently in the case of accuracy in the 0.2$\times$ LR setting; see Figure 19b); and (2) the $n_{\mathcal{D}}$ which results in the best performance is different – while $n_{\mathcal{D}} = 50$ leads to the best results for MNIST at $\varepsilon = 10$ in Section 4, $n_{\mathcal{D}} = 200$ performs the best for 5$\times$ LR and $n_{\mathcal{D}} = 100$ is

the best for $0.2\times$ LR. Note that these two differences are *not observed* in the experiments where we vary discriminator size $d_{\mathcal{D}}$ in Appendix D.1: all runs with different $d_{\mathcal{D}}$ track the $d_{\mathcal{D}} = 128$ run closely.

In Figures 20 and 21, we examine the case where we only adjust $\mathcal{D}$ LR, by $5\times$ and $0.2\times$ respectively, and keep $\mathcal{G}$ LR fixed at 0.0002. Again, we observe large improvements in utility as we take $n_{\mathcal{D}} >> 1$, up to the point where $n_{\mathcal{D}}$ is too high. We note that when keeping $\mathcal{G}$ LR fixed, the $0.2\times$ setting gets much closer to the level of improvement from varying $n_{\mathcal{D}}$ observed in the base LR setting.

In summary: changing learning rates while keeping other hyperparameters fixed still exhibits the benefit of increasing $n_{\mathcal{D}}$, but compared to the base setting, does not recover: (1) the scale of the improvement, and (2) the precise behaviour of the phenomenon; i.e. the same optimal $n_{\mathcal{D}}$. We leave open the question of understanding more precisely how the phenomenon changes under different learning rates: it may be fruitful to investigate how Adam's momentum parameters $(\beta_1, \beta_2)$ and DPSGD noise level $\sigma$ impact the results, and also perhaps the degradation of the non-private GAN results for large $\mathcal{D}$ LR.

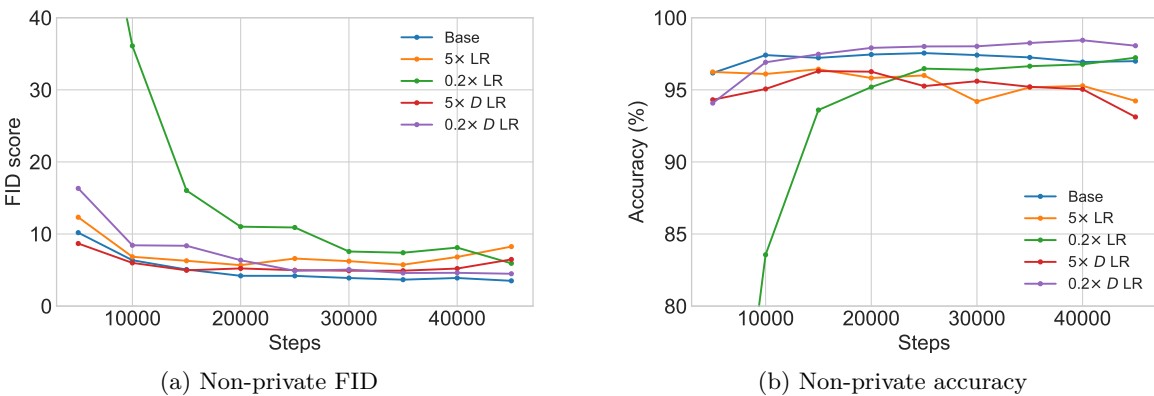

(a) Non-private FID

(b) Non-private accuracy

Figure 17: MNIST FID and downstream classifier accuracy for non-private GAN training with various learning rate settings. The blue ($d_{\mathcal{D}} = 128$) line corresponds to the base learning rate setting used in previous experiments.

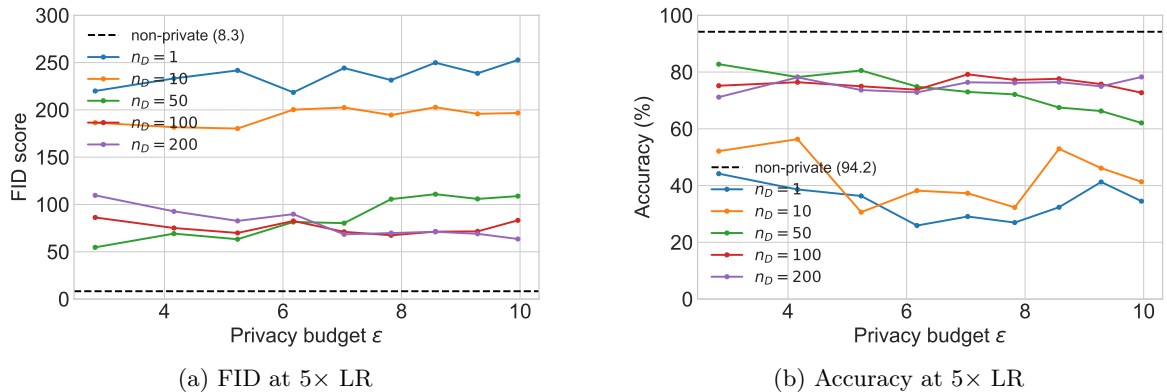

(a) FID at $5\times$ LR

(b) Accuracy at $5\times$ LR

Figure 18: MNIST FID and downstream classifier accuracy for DPGAN training runs targeting $\varepsilon = 10$ and using $5\times$ the base learning rate for both $\mathcal{G}$ and $\mathcal{D}$, under various settings of $n_{\mathcal{D}}$.

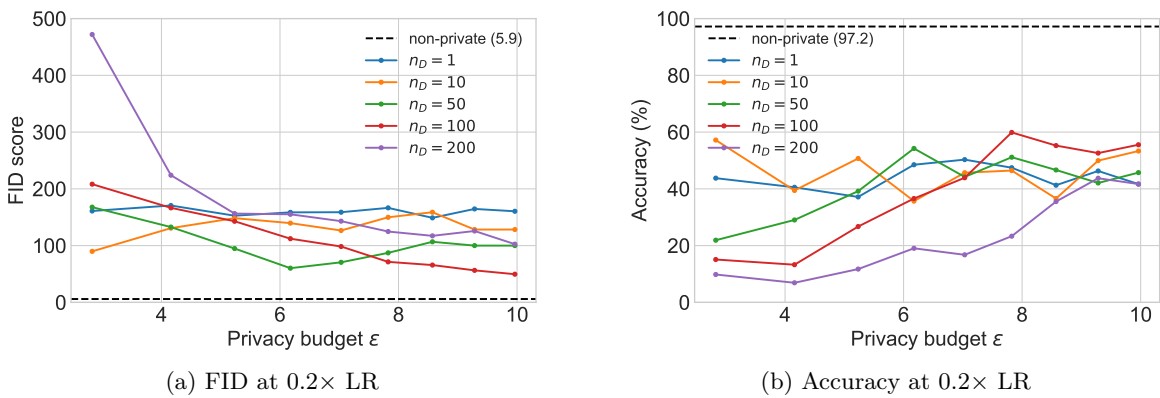

(a) FID at 0.2× LR

(b) Accuracy at 0.2× LR

Figure 19: MNIST FID and downstream classifier accuracy for DPGAN training runs targeting $\varepsilon = 10$ and using $0.2\times$ the base learning rate for both $\mathcal{G}$ and $\mathcal{D}$, under various settings of $n_{\mathcal{D}}$.

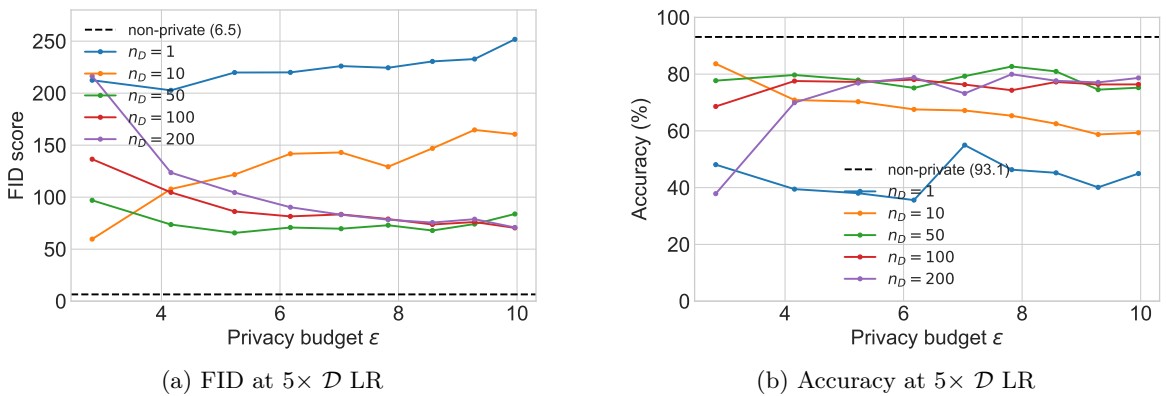

(a) FID at 5× $\mathcal{D}$ LR

(b) Accuracy at 5× $\mathcal{D}$ LR

Figure 20: MNIST FID and downstream classifier accuracy for DPGAN training runs targeting $\varepsilon = 10$ and using $5\times$ the base learning rate for $\mathcal{D}$ only ($\mathcal{G}$ LR unchanged), under various settings of $n_{\mathcal{D}}$.

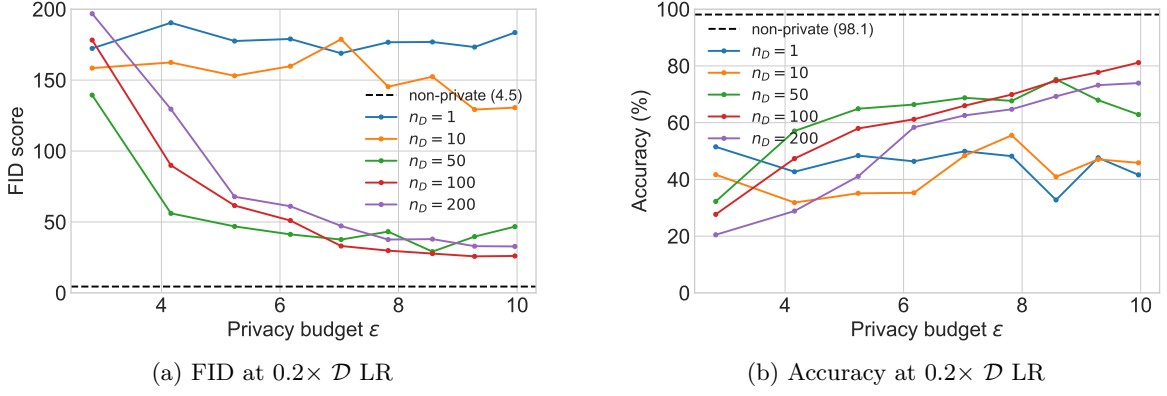

(a) FID at 0.2× $\mathcal{D}$ LR

(b) Accuracy at 0.2× $\mathcal{D}$ LR

Figure 21: MNIST FID and downstream classifier accuracy for DPGAN training runs targeting $\varepsilon = 10$ and using $0.2\times$ the base learning rate for $\mathcal{D}$ only ($\mathcal{G}$ LR unchanged), under various settings of $n_{\mathcal{D}}$.

### D.3 Varying batch size and noise level

Fixing a batch size $B$ and a noise level $\sigma$ yields a total discriminator step budget $T$ allowed under our privacy budget $\varepsilon$. For example, the results from Section 4 and Appendices D.1 and D.2 use $B = 128$ and $\sigma = 1$, which allows for $T = 450\text{K}$ when targeting $\varepsilon = 10$ on MNIST. Again targeting MNIST at $\varepsilon = 10$, we take

various combinations of $(B, \sigma)$, and plot the final FID and accuracy of DPGANs trained at such a setting, over a spectrum of $n_\mathcal{D}$. Results are pictured in Figures 22, 23, and 24.

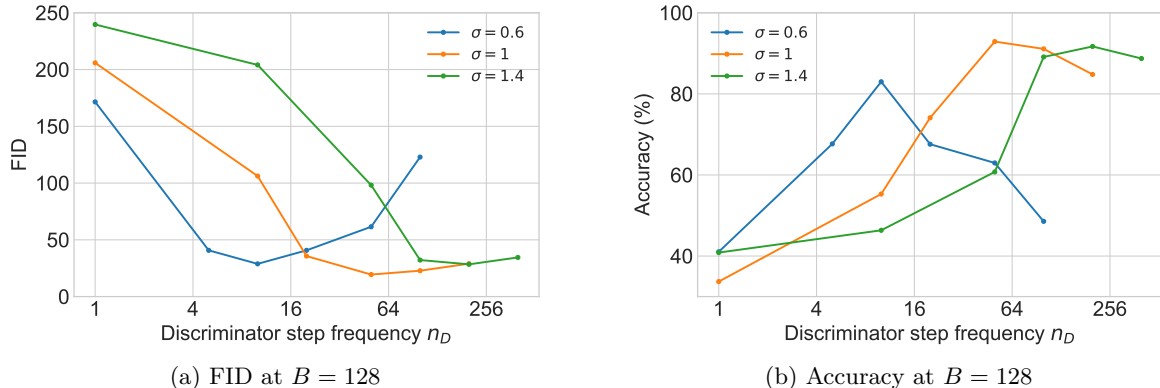

(a) FID at $B = 128$        (b) Accuracy at $B = 128$

Figure 22: MNIST FID and downstream classifier accuracy for $B = 128$ runs targeting $\varepsilon = 10$, with $\sigma \in \{0.6, 1, 1.4\}$. We report final utility over a range of $n_\mathcal{D}$ for the 3 noise levels. The x-axis is log-scaled.

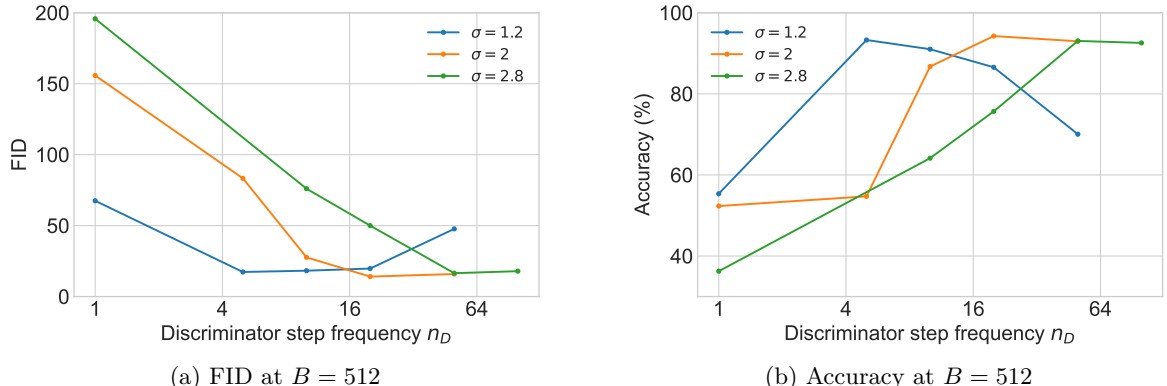

(a) FID at $B = 512$        (b) Accuracy at $B = 512$

Figure 23: MNIST FID and downstream classifier accuracy for $B = 512$ runs targeting $\varepsilon = 10$, with $\sigma \in \{1.2, 2, 2.8\}$. We report final utility over a range of $n_\mathcal{D}$ for the 3 noise levels. The x-axis is log-scaled.

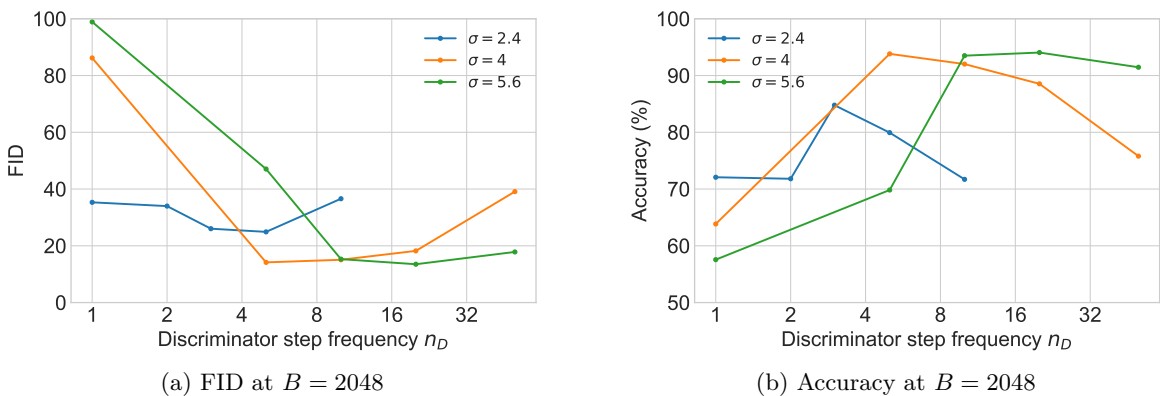

(a) FID at $B = 2048$        (b) Accuracy at $B = 2048$

Figure 24: MNIST FID and downstream classifier accuracy for $B = 2048$ runs targeting $\varepsilon = 10$, with $\sigma \in \{2.4, 4, 5.6\}$. We report final utility over a range of $n_\mathcal{D}$ for the 3 noise levels. The x-axis is log-scaled.

**Results.** At all batch sizes and noise levels, we observe the same U-shaped utility curve described in Section 4, which predicts the existence of an optimal $n_\mathcal{D}$ for any fixed setting of $(\sigma, B)$. For fixed $B$, the

optimal $n_{\mathcal{D}}$ is lower for smaller $\sigma$. We also see that for settings with low $\sigma$ and large $B$, optimal $n_{\mathcal{D}}$ can be quite low. For all batch sizes, choosing noise levels that achieve their optimal $n_{\mathcal{D}}$ at fairly large values $(>> 1)$ tends to outperform smaller noise levels which achieve their optimal $n_{\mathcal{D}}$ early.

## E   Additional results

### E.1   Wall clock times

We report wall clock times for training runs under various hyperparameter settings, which are executed on $1\times$ NVIDIA A40 card setups running PyTorch 1.11.0+CUDA 11.3.1 and Opacus 1.1.3. Table 6 presents results on MNIST, in particular comparing the effect of $n_{\mathcal{D}}$ on training time. The total number of discriminator steps, $T$, is determined by the privacy budget and DPSGD hyperparameters. Hence, increasing $n_{\mathcal{D}}$ results in fewer total $\mathcal{G}$ steps and faster training time. Table 7 presents training times under adaptive discriminator step frequency for various datasets.

All private settings are much slower than non-private training. Indeed, the best DP results tend to come from training long with large noise levels, trading off computation for utility (De et al., 2022). For example, the best DP diffusion models (Dockhorn et al., 2022) use 8 V100's for 1 day to train their best MNIST models. Although not directly comparable, we note that our best $\varepsilon = 10$ results train in 7.5 hours on 1 A40.

| Privacy | $B$ | $\sigma$ | $T$ | $n_{\mathcal{D}}$ | FID | Wall clock time |
|---|---|---|---|---|---|---|
| $\varepsilon = \infty$ | 128 | - | 45K | 1 | $3.4 \pm 0.1$ | 44m |
| $\varepsilon = 10$ | 128 | 1 | 450K | 1 | $205.3 \pm 0.9$ | 11h 03m |
| | | | | 10 | $103.4 \pm 5.8$ | 6h 33m |
| | | | | 50 | $18.5 \pm 0.9$ | 5h 56m |
| | | | | 100 | $21.0 \pm 1.6$ | 5h 57m |
| | | | | 200 | $26.6 \pm 2.2$ | 5h 54m |
| | 2048 | 5.6 | 98K | 20 | $13.2 \pm 1.0$ | 16h 54m |
| $\varepsilon = 1$ | 128 | 5 | 325K | 200 | $111.1 \pm 17.9$ | 4h 40m |
| | 512 | 14 | 165K | 50 | $106.2 \pm 64.0$ | 7h 31m |

Table 6: Wall clock times on MNIST for various settings. The privacy level $\varepsilon$, batch size $B$, and noise level $\sigma$ determines the total number $\mathcal{D}$ steps taken during training, $T$. Given $T$, the discriminator update frequency $n_{\mathcal{D}}$ determines the number of $\mathcal{G}$ steps taken during training.

| Privacy | Dataset | $B$ | $\sigma$ | $T$ | FID | Wall clock time |
|---|---|---|---|---|---|---|
| $\varepsilon = \infty$ | MNIST | 128 | - | 45K | $3.4 \pm 0.1$ | 44m |
| | FashionMNIST | | | | $16.5 \pm 1.7$ | 42m |
| | CelebA | | | | $30.0 \pm 1.6$ | 47m |
| $\varepsilon = 10$ | MNIST | 512 | 2 | 174K | $12.8 \pm 0.3$ | 7h 35m |
| | FashionMNIST | | | | $62.3 \pm 8.7$ | 7h 35m |
| | CelebA | 2048 | 4 | 385K | $170.8 \pm 20.3$ | 3d 17h 49m |
| $\varepsilon = 1$ | MNIST | 512 | 14 | 165K | $52.6 \pm 3.2$ | 7h 43m |
| | FashionMNIST | | | | $126.4 \pm 4.1$ | 7h 26m |

Table 7: Wall clock times on MNIST, FashionMNIST and 32$\times$32 CelebA for runs using adaptive discriminator step frequency (with the exception of the $\varepsilon = \infty$ results, which use $n_{\mathcal{D}} = 1$).

### E.2   Increasing discriminator learning rate instead of step frequency

From the experimental setting in Section 4 targeting MNIST at $\varepsilon = 10$, we consider the $n_{\mathcal{D}} = 1$ setting, and increase the discriminator learning rate instead of $n_{\mathcal{D}}$. $\mathcal{G}$ LR is kept fixed at 0.0002. Results are in Table

8. We do not observe the same level of improvement obtained by increasing $n_{\mathcal{D}}$ and keeping $\mathcal{D}$ LR/$\mathcal{G}$ LR at $1\times$.

| $\mathcal{D}$ LR/$\mathcal{G}$ LR | FID | Acc. (%) |
|:---:|:---:|:---:|
| $1\times$ | 205.9 | 33.7 |
| $5\times$ | 251.8 | 45.0 |
| $10\times$ | 228.1 | 37.5 |
| $50\times$ | 237.2 | 54.5 |

Table 8: Results on MNIST at $\varepsilon = 10$ for increasing $\mathcal{D}$ LR while keeping $n_{\mathcal{D}} = 1$. For reference, using $n_{\mathcal{D}} = 50$ while keeping $\mathcal{D}$ LR/$\mathcal{G}$ at $1\times$ yields an FID score of $18.5 \pm 0.9$ and accuracy of $93.0 \pm 0.6\%$.

## F   Additional discussion

**DP tabular data synthesis.** Our investigation focuses on image datasets, while many important applications of private data generation involve tabular data. In these settings, marginal-based approaches (Hardt et al., 2012; Zhang et al., 2017; McKenna et al., 2019) perform the best. While Tao et al. (2021) find that private GAN-based approaches fail to preserve even basic statistics in these settings, we speculate that our techniques may yield improvements.

**Taking multiple discriminator steps.** The original GAN formulation of Goodfellow et al. (2014) has $n_{\mathcal{D}}$, the number of discriminator steps taken between generator steps, as a tunable hyperparameter. In the WGAN framework (Arjovsky et al., 2017), it is suggested to train discriminators as much as possible between generator steps, i.e. to optimality, for best performance. In practice, WGAN implementations use $n_{\mathcal{D}} = 5$ to save on computation. Several studies empirically explore the effect of taking multiple discriminator steps (Miyato et al., 2018; Brock et al., 2019), finding that searching for an optimal $n_{\mathcal{D}}$ can improve results. Similar imbalanced setups, such as lookahead and imbalanced learning rates, have been analyzed theoretically (Chavdarova et al., 2021; Fiez & Ratliff, 2021).

In the private setting, applying such strategies to improve DPGAN training has been relatively unexplored. DPGAN training recipes are largely ports of non-private approaches – inheriting many parameter choices designed for performant non-private training which are sub-optimal under DP.

Guidance in the non-private setting (tip 14 of Chintala et al. (2016)) prescribes to train the discriminator for more steps in the presence of noise (a regularization approach used in non-private GANs). This is the case for DP, and is our core strategy that yields the most significant gains in utility. We were not aware of this tip when we discovered this phenomenon, but it serves as validation of our finding. While Chintala et al. (2016) provides little elaboration, looking at further explorations of this principle (and other strategies) in the non-private setting may offer guidance for improving DPGANs. For instance, Chavdarova et al. (2021) propose a lookahead update rule that enables fast convergence in the presence of noise, without using large batches – such techniques may help in the private setting as well.

**Hyperparameter tuning in DP machine learning.** Hyperparameter tuning is crucial to getting deep learning to work. The same is true under privacy, with two additional concerns: (1) tuning is not free – naive composition says privacy loss scales with the number of runs; and (2) DPSGD alters the hyperparameter landscape – introducing extra ones, and also *changing the relative importance of existing hyperparameters*. (1) is addressed by Liu & Talwar (2019) (although composition is competitive in settings where adaptive selection is important (Mohapatra et al., 2022)). On (2), Ponomareva et al. (2023) offers a comprehensive guide on DP hyperparameter tuning; for GAN training, our work identifies an important parameter with outsized impact in the DP setting.

To compare with prior work, we report our best hyperparameter settings. Indeed, introducing a highly dataset-dependent parameter can result in worse performance overall when accounting for the cost of search in a real deployment setting. Our adaptive discriminator update frequency is motivated by this concern, and our use of MNIST hyperparameters directly for FashionMNIST is a brittleness sanity-check.

The aspect of our work that identifies the importance of discriminator update frequency in private GAN training is unaffected by concerns regarding private hyperparameter search.

Evaluation approaches that take into account the cost of search when comparing algorithms is an important direction, which we do not address in this work. For benchmark datasets, the problem is complicated by implicit knowledge encoded in various algorithms' design choices and "default" hyperparameter ranges.

