# OpenReview forum: "Private GANs, Revisited"
_TMLR — Accepted by TMLR_

### Review · Reviewer_ZVUN · 2023-04-01

**Summary Of Contributions:**

This paper proposes a training algorithm for private GANs, which significantly improves existing methods on private GANs. In private GANs, the discriminator is updated using langevin-type stochastic gradients (clipped stochastic gradient with additive Gaussian noise), resulting in a potential weak discriminator. Such an observation motivates the algorithm in the paper, which targets at "balancing" the effective training of generator and discriminator. In particular, the discriminator is trained for multiple iterations until the generator is updated once. This method is intuitive and relatively simple, yet this should not be perceived as a weakness. In fact, the performance improvement of the proposed algorithm is statistically significant compared to methods in the same class (except diffusion models). Therefore, the empirical evaluation justifies the contribution of the paper.

**Audience:**

Yes

**Broader Impact Concerns:**

No concerns.

**Claims And Evidence:**

Yes

**Requested Changes:**

Definition 1 and Proposition 2 are slightly distracting. I understand the intention of a rigorous exposure, however, these results are not needed to appreciate the strength of the paper.

Mode collapse in Figure 3 needs a better explanation. Looking at the figures, I suspect 7 and 9 are mixed and similar.

It appears that the discriminative power of discriminator is highly relevant to the performance of DPGANs. Does varying the size of discriminator influences the performance?

**Strengths And Weaknesses:**

Authors keep a clean writing style and the paper is easy to follow. I appreciate a comprehensive experimental study of allowing multiple discriminator steps for the discriminator. In addition to support the superior performance of the proposed algorithm, there are plenty of discussions regarding the tradeoff on the number of multiple discriminator steps, mode collapse, and adaptive discriminator steps.

As mentioned by authors, the performance of DPGAN does not match that of diffusion models. This could undermine the value of the paper. However, GAN often leads to fast generation due to the easy generator structure, while diffusion models are slow in generation because of the iterative backward simulation. In order to strengthen the result, it might be helpful to argue that DPGANs generate much faster with competitive performance to diffusion models. (Minor) The generated images in Figure 8 also Figure 13 seem not very appealing compared to diffusion models.

---

> ### Author Response · Authors · 2023-07-06
> **Reponse to Reviewer ZVUN**
>
> Thank you for the thoughtful comments regarding our work.
>
> We are glad the reviewer finds our empirical results significant, and appreciates the simple and intuitive nature of our proposed solution. We also appreciate that the reviewer finds our experiments and discussions illustrating the phenomenon to be comprehensive.
>
> To take into account feedback, we’ve updated the manuscript with changes to writing and also added extra results.
>
> In the following, we reply to specific points raised by the reviewer.
>
> ***The performance of DPGAN does not match that of diffusion models***
>
> Indeed, our DPGAN results do not match the results of DP diffusion models. As the reviewer points out, there are reasons to prefer a GAN over diffusion models: i.e. faster inference time and controllable latent spaces. Recent work on text-to-image for GANs (Kang et al., 2023) has shown that GANs can be scaled up effectively. Furthermore, a significant aspect of our work is about correcting the misconception in the DP synthetic literature data that there are inherent problems with the DPGAN framework, rather, they were improperly trained.
>
> We also note that the best DP diffusion models were trained on significantly higher compute budgets: Dockhorn et al. (2022) use 8 V100’s for 1 day to train their best MNIST model, while our best $\varepsilon=10$ results train in 7.5 hours on a single A40. With the caveat that they are not directly comparable, since there are differences in the setup (although A40s are roughly as fast as V100s), our GANs are trained with significantly fewer GPU hours.
>
> ***Requested change: Definition 1 and Proposition 2 are slightly distracting***
>
> Point taken. Since we are not proving any theorems, we agree the extra notation introduced with Proposition 2 is unnecessary. We’ve replaced it with an informal description of the post-processing property of DP, which we refer to in our explanation of the privacy guarantee of the DPGAN training algorithm. For completeness, we prefer to keep Definition 1.
>
> ***Requested change: Mode collapse in Figure 3 needs a better explanation***
>
> We’ve updated the explanation. The main point we wanted to make is that training longer (and expending more privacy budget) caused intra-digit similarity to increase, not capturing the full variation present in the underlying data distribution.
>
> ***Requested change: Does varying the size of discriminator influences the performance?***
>
> It’s an interesting question. We’ve added this extra result (please see Appendix D.1). Keeping everything else fixed, and varying the number of filters in the discriminator (leading to discriminators with roughly 0.25x, 0.5x, and 2x the parameter count), we report MNIST @ $\varepsilon=10$ results using various $n_{\mathcal D}$. Results track the 1x parameter count results quite closely. We see slight performance degradations for the smallest discriminators, but at the same time, compute savings.
>
> —--------------------------------
>
> We thank the reviewer for their suggestions and for engaging with our work. We hope that our comments here and our revisions helped address the reviewer’s concerns.

---

> > ### Author Response · Authors · 2023-07-06
> > **References**
> >
> > **References**
> >
> > Tim Dockhorn, Tianshi Cao, Arash Vahdat, and Karsten Kreis. Differentially private diffusion models. *CoRR*, abs/2210.09929, 2022.
> >
> > Minguk Kang, Jun-Yan Zhu, Richard Zhang, Jaesik Park, Eli Shechtman, Sylvain Paris, and Taesung Park. Scaling up GANs for text-to-image synthesis. In *Proceedings of the 2023 IEEE Conference on Computer Vision and Pattern Recognition (CVPR’23)*, 2023.

---

### Review · Reviewer_sgjK · 2023-05-07

**Summary Of Contributions:**

The paper studies differentially private training of generative adversarial networks (GAN). The authors show that for a given hyperparameter configuration, taking more discriminator steps per generator step may improve performance (though there's a point at which further increasing leads to performance degradation). Authors also study additional modifications such as large batch training and adaptive discriminator step frequency and show further improvements in results.

**Audience:**

Yes

**Claims And Evidence:**

No

**Requested Changes:**

Those listed in the "Weaknesses" section.

**Strengths And Weaknesses:**

**Strengths**

The authors revisit differentially private training of GANs and perform a more detail-oriented empirical analysis of this framework. By carefully understanding how different design choices of training affect performance, the authors show that better performance can be obtained.

**Weaknesses**

- The authors claim to have proposed the scheme of taking more discriminator steps per generator step. However, this imbalanced stepping scheme was first presented in the original GAN paper and prominent follow-up papers such as WGAN [1]. To be clear, the cite references (Xie et al. 2018) already employed the imbalanced strategy (5 D steps per G step). Authors should clarify the present work's relationship with prior works. The present work (at least for the content related to imbalanced stepping) feels more like an empirical analysis of an existing technique than proposing a new technique.
- Authors claim that increasing the D steps per G step can lead to improved performance and justify this claim by showing experiments with other hyperparameters fixed. However, authors only present results for one set of fixed (other) hyperparameters. For a more compelling argument, authors ideally should repeat the same set of experiments (e.g., those in Figure 2), varying other hyperparameters such as the learning rate of the two optimizers to show that the phenomenon is robust to changes in the overall setup.
- Authors claim that the technique is general but only perform experiments for image generation. To make a more compelling argument, authors should either clarify that the scope is image generation or present additional results for other modalities of data (e.g. tabular data generation).

[1] Arjovsky, Martin, Soumith Chintala, and Léon Bottou. "Wasserstein generative adversarial networks." International conference on machine learning. PMLR, 2017.

**Side notes**
- Have authors explored other loss functions (e.g., those in WGAN paper)?

---

> ### Author Response · Authors · 2023-07-06
> **Response to Reviewer sgjK**
>
> Thank you for the thoughtful comments on our work.
>
> To take into account feedback, we’ve updated the manuscript with changes to writing and also added extra results.
>
> In the following, we reply to specific points raised by the reviewer.
>
> ***Requested change: Clarifying connection with prior works, especially regarding the origin of imbalanced stepping***
>
> We have revised the paper with an extended discussion of prior work, and have clarified our claims.
>
> Indeed, as pointed out by the reviewer, imbalanced stepping was in the seminal study of Goodfellow et al. (2014), as well as in WGAN (Arjovsky et al., 2017). In both, $n_{\mathcal D}$ is presented as a hyperparameter. Xie et al. (2018) follows the framework of WGAN, and takes 5 $\mathcal D$ steps per $\mathcal G$ step. We have revised the draft to make this more evident.
>
> Based on our results in Figure 1, the choice of $n_{\mathcal D}$ leads to dramatically different results: it is the difference between SOTA-competitive results and something that is entirely not working. However Xie et al. (2018) (as well as Torkzadehmahani et al. (2019),  which uses $n_{\mathcal D}=1$, and Long et al. (2021); Chen et al. (2020); Wang et al. (2021), which reproduce DPGAN results as a baseline) makes no mention of the importance of the parameter, nor the idea that for a given noise level and privacy budget, there is a range of $n_\mathcal D$ that allows for successful GAN training (see Appendix D.3).
>
> Hence we believe the imbalanced stepping in Xie et al. (2018) is the consequence of directly porting over non-private WGAN settings – the study does not understand it as a key aspect of adapting non-private GAN training recipes to DP.
>
> Therefore we disagree with the assertion that our work is an empirical analysis of an existing technique – the counterfactual being that if the role of $n_{\mathcal D}$ was known prior: (a) it would have been mentioned in one of these many papers that train DPGANs, since setting it properly is crucial to successful training; and (b) they would have found that proposed alternative GAN privatization schemes are outperformed by DPGAN.
>
> ***Requested change: Ablations***
>
> We’ve added ablation results showing how the phenomenon responds to different setups. Please see Appendix D. In these experiments, we target the setting of $\varepsilon=10$ on MNIST and examine the effect of increasing $n_{\mathcal D}$, after straying from the base setup in:
>
> 1. Discriminator size
> 2. Discriminator and Generator learning rates
> 3. Batch size and noise level
>
> To summarize our findings: in all cases, we observe significant performance improvements due to increased $n_{\mathcal D}$. However, based on the particular variation from the base setup, the magnitude, as well as precise behaviour (i.e. the optimal $n_{\mathcal D}$) of the effect of larger $n_{\mathcal D}$ varies. For example, in the learning rate experiments, we find that accuracy and FID achieved by the optimal $n_{\mathcal D}$ is worse than in our base learning rate setting, and the optimal $n_\mathcal{D}$ is different. In our batch size and noise level experiments, the optimal $n_\mathcal{D}$ shifts, and most settings exhibit the degree of improvement from increasing $n_{\mathcal D}$ as in the base setting. Our experiments with varied discriminator size tracked our original setting very closely.
>
> ***Requested change: Clarify that the scope is image generation***
>
> We’ve clarified the scope of the paper to be about image generation. Our speculative comments on possible relevance to tabular data generation have been moved to the appendix.
>
> ***Have authors explored other loss functions (e.g., those in WGAN paper)?***
>
> We have not. All experiments are carried out with non-saturating GAN loss from Goodfellow et al. (2014).
>
> —--------------------------------
>
> We thank the reviewer for their suggestions and for engaging with our work. We hope that our comments here and our revisions helped address the reviewer’s concerns.

---

> > ### Author Response · Authors · 2023-07-06
> > **References**
> >
> > **References**
> >
> > Ian Goodfellow, Jean Pouget-Abadie, Mehdi Mirza, Bing Xu, David Warde-Farley, Sherjil Ozair, Aaron Courville, and Yoshua Bengio. Generative adversarial nets. In *Advances in Neural Information Processing Systems 27 (NIPS’14)*, 2014.
> >
> > Martin Arjovsky, Soumith Chintala, and Léon Bottou. Wasserstein generative adversarial networks. In *Proceedings of the 34th International Conference on Machine Learning (ICML’17)*, 2017.
> >
> > Liyang Xie, Kaixiang Lin, Shu Wang, Fei Wang, and Jiayu Zhou. Differentially private generative adversarial network. *CoRR*, abs/1802.06739, 2018.
> >
> > Reihaneh Torkzadehmahani, Peter Kairouz, and Benedict Paten. DP-CGAN: Differentially private synthetic data and label generation. In *IEEE Conference on Computer Vision and Pattern Recognition Workshops, (CVPR Workshops’19)*, 2019.
> >
> > Yunhui Long, Boxin Wang, Zhuolin Yang, Bhavya Kailkhura, Aston Zhang, Carl Gunter, and Bo Li. G-PATE: Scalable differentially private data generator via private aggregation of teacher discriminators. In *Advances in Neural Information Processing Systems 34 (NeurIPS’21)*, 2021.
> >
> > Dingfan Chen, Tribhuvanesh Orekondy, and Mario Fritz. GS-WGAN: A gradient-sanitized approach for learning differentially private generators. In *Advances in Neural Information Processing Systems 33 (NeurIPS’20)*, 2020.
> >
> > Boxin Wang, Fan Wu, Yunhui Long, Luka Rimanic, Ce Zhang, and Bo Li. DataLens: Scalable privacy preserving training via gradient compression and aggregation. In *CCS’21: 2021 ACM SIGSAC Conference on Computer and Communications Security*, 2021.

---

### Review · Reviewer_UGFA · 2023-06-02

**Summary Of Contributions:**

This paper focuses on improving existing differentially private (DP) GAN models.
It documents that using a different update ratio---that is, updating the discriminator multiple times versus the generator---significantly improves the quality of the generated samples.


My main concern is the limited novelty of this paper: in particular, the change relative to prior work in the DP context, is that the discriminator is trained multiple times, that is, only line 10 in Alg. 1.
Moreover, it is well-known in the GAN community that multiple discriminator updates help significantly the optimization, and almost all publicly available GAN implementations use this; see references below.
Moreover, given that the paper has empirical contributions, these are not thoroughly carried out (with seeds and standard deviation, reporting performances over wall clock time or gradient queries, an additional metric that the generated data is differentially private, etc.; see below), and the question on how to select the update ratio in this DP context remains open.

**Audience:**

Yes

**Broader Impact Concerns:**

/

**Claims And Evidence:**

Yes

**Requested Changes:**


**Experiments.**
- Please report the standard deviation over at least 3 seeds.
- Please depict a plot where the x-axis is wall clock time or gradient queries.

**Motivation.**
Please explain better what is the overall motivation: it is surprising to me that the overall goal is not to use


## Questions:
1. Increasing $n_D$ also increases the computation a lot. Have you tried increasing the step size for the discriminator?
2. What would be a recommended way to select $n_D$ after this study?



## Minor / Typos
- Abstract: it would be helpful to make the contributions more precise. For example:
   - "after modifications to training" -- it would be clearer to list these
   - "a careful balance between the generator and discriminator" -- it is unclear what this means, balance in what?
   - "restores parity" -- it is unclear what this means
   - "we experiment with other modifications" -- these should be listed
   - "improvements in generation quality" -- give a precise relative improvement in FID.
- Sec 1: "weakens the discriminator relative to the generator"--while I understand the authors' point, this is very informal phrasing; moreover, since this part is the main motivation for the manuscript, it may be helpful to define things or alternatively to explain something like "X happens, thus herein we refer to it as Y".
- Sec 1: "Disrupting the careful balance necessary for successful GAN training" -- this is unclear. What is meant by careful balance?
- Sec 1: similarly, "addresses the imbalance introduced by noise"
- Sec 1, first contribution: do you mean "outperform" instead of "compete"?
- Sec 1, third contribution: it is unclear how the third contribution differs from the previous two contributions



**Strengths And Weaknesses:**

# Strengths

This paper improves the performances of previous implementations of DP GANs, by using multiple discriminator updates per one generator update. It also studies if larger batch sizes and adaptive discriminator steps improve the performances for such training with multiple discriminator updates.



# Weaknesses

**Novely/Relevance/Contribution.**
This fact that multiple updates for the discriminator help is a well-known observation in the GAN community, both as a theoretical and empirical observation, see for example [1-4]. It is related to using larger step sizes for the discriminator (instead of updating it multiple times). Thus, I am surprised that the first differentially private GAN did not use this because all standard GAN setups do that.
That said, this paper tries to interpret that using an argument based on "asymmetric noise addition", which is only argued, but does not provide theoretical insights.
Given that this fact is well-known (from an optimization perspective), this perspective of asymmetric noise addition should be more thoroughly carried out for it to offer a different/deeper understanding in the DP context.

**Misleading discussion on batch sizes.**
It is known that noisy gradient estimates can "harm" the optimization of GANs [5], and this is true for any min-max optimization, thus also for DP training.
Now, for simple datasets such as MNIST, there is no/little gain in increasing the batch size, but for more complex datasets such as ImageNet, this is necessary in order to ensure that the optimization method does not diverge away rapidly [6].
Now, this paper notes that for non-private GAN on MNIST, increasing the batch size does not help, but for private GAN, where noise is added, it does help.
This is very much expected/well-known because the added noise increased the complexity of the dataset, and from an optimization perspective, controlling for the variance of the noisy gradient estimates helps the convergence.
Thus, the discussion that this is something specific to DP GANs is in my opinion very misleading  (and the conclusion that for non-private GANs increased batch size does not help is incorrect).


**Writing.**
The writing is often informal, which makes the reading ambiguous. For example, it is often used "striking a balance", or "careful balance" without being clear about what that means; see further examples below.


**Experiments.**
The experiments do not report standard deviation. Moreover, they do not take into account the added computational expense since now, at each generator step, there are $n_D$ more gradient queries, and this number can be large.
Furthermore, it is unconvincing to me that FID is a good measure since the primary goal is to have a differentially private generative model, and FID reports how similar the generative samples are to the dataset samples.


### Refs
- [1] Takeru Miyato, Toshiki Kataoka, Masanori Koyama, and Yuichi Yoshida. Spectral normalization for generative adversarial networks. In ICLR, 2018.
- [2] Gauthier Gidel, Hugo Berard, Gaetan Vignoud, Pascal Vincent, Simon Lacoste-Julien, A Variational Inequality Perspective on Generative Adversarial Networks, In ICLR, 2019.
- [3] Tatjana Chavdarova, Matteo Pagliardini, Sebastian U Stich, François Fleuret, and Martin Jaggi. Taming GANs with Lookahead-Minmax. In ICLR, 2021.
- [4]  Tanner Fiez, Lillian J Ratliff. Local Convergence Analysis of Gradient Descent Ascent with Finite Timescale Separation. In ICLR, 2021.
- [5] T. Chavdarova, G. Gidel, F. Fleuret, and Simon Lacoste-Julien. Reducing noise in GAN training with variance reduced extragradient. In NeurIPS, 2019.
- [6] A. Brock, J. Donahue, and K. Simonyan. Large-scale GAN training for high-fidelity natural image
synthesis. In ICLR, 2019.

---

> ### Author Response · Authors · 2023-06-09
> **Thanks!**
>
> Thank you for your thorough review! We are working on a response and revision with these comments in mind.
>
> Would you mind please completing the following sentence from your review, so we could appropriately respond to it? Thanks!
> "Motivation. Please explain better what is the overall motivation: it is surprising to me that the overall goal is not to use"

---

> ### Author Response · Authors · 2023-07-06
> **Response to Reviewer UGFA**
>
> Thanks for the thoughtful comments on our work.
>
> To take into account feedback, we’ve updated the manuscript with changes to writing and also added extra results.
>
> In the following, we reply to specific points raised by the reviewer.
>
> ***On novelty/relevance/contribution: ‘multiple updates help’ is known in the GAN community; surprising that DPGANs did not use it; lack of theoretical results.***
>
> Indeed, in the *non-private* setting, multiple discriminator steps go all the way back to the original formulation of GANs in Goodfellow et al. (2014), where it is presented as a tunable hyperparameter. We appreciate the extra references for works that study the effect both empirically and theoretically, and have included them in our discussion (see Appendix F).
>
> In contrast, in the *private setting*, the effect of this hyperparameter $n_\mathcal{D}$ has been significantly underexplored. We suggest the following explanation for why: the mentioned papers studying the effect explore low values $\leq5$ (Miyato et al., 2018; Gidel et al., 2019; Brock et al., 2019) – and as GAN architectures co-evolved with training recipes, several of the most popular works opted to eliminate $n_\mathcal{D}$ from the design space, un-emphasizing its choice and using a small fixed number like 5 (WGAN (Arjovsky et al., 2017)) or 1 (DCGAN (Radford et al., 2016) and StyleGAN(2|3|XL) (Karras et al., 2019; 2020; 2021; Sauer et al., 2022)). These setups and parameter choices were designed for performant non-private training – a crucial point missed when GANs were adapted to DP and many parameter choices were left untouched. In particular, the work of Xie et al. (2018), which privatizes WGAN, uses $n_{\mathcal D}=5$ from WGAN.  Torkzadehmahani et al. (2019) privatizes DCGAN, and keeps $n_{\mathcal D}=1$. Both use small batch sizes.
>
> The primary contribution of our work is revisiting these design choices that have become unemphasized in the non-private world, showing that they make dramatic differences in the DP setting. Such modifications may appear obvious to a GAN expert; for example, Tip 14 from GANhacks (Chintala et al., 2016) suggests taking extra $\mathcal D$ steps when noise is added to the discriminator, which is precisely what we do. However in the DP synthetic data literature, these results are highly novel: 5 recent methods for DP synthetic data (GPATE (Long et al., 2021), GS-WGAN (Chen et al., 2020), DP Sinkhorn (Cao et al., 2021), DP-Hermite (Vinaroz et al., 2022), DataLens (Wang et al., 2021)) published at {NeurIPS, ICML, CCS} report weak DPGAN baselines (which in some cases, serve as the motivation for their approach). Our results thus expose a common failure in terms of baseline comparisons across a broad range of results on private generative models. With proper tuning as in our paper, we outperform prior DPGAN baselines, and in many cases, proposed approaches in these papers.
>
> Our study is empirical. We propose an explanation consistent with our results: performance improvement requires sufficiently high discriminator accuracy before updating the generator, DP makes this difficult, and more steps remedies the problem. We do not perform any theoretical analysis, which is an interesting avenue for future work. One caveat with coming up with a good theory for this phenomenon is a known challenge in the non-private theory of GANs. Jelassi et al. (2022) runs an experiment (see Figure 1 in their study) targeting the setting of aforementioned work of Fiez & Ratliff (2020), where the ratio between D LR and G LR is varied. They find that convergence to local minimax points and GAN performance are decorrelated: while large discriminator to generator learning rate ratios improve convergence, they result in worse FID. Out of the box, theoretical results from the optimization perspective of the form “larger $n_\mathcal{D}$ improves convergence” does not explain our results.

---

> > ### Author Response · Authors · 2023-07-06
> > **Response to Reviewer UFGA, cont.**
> >
> > ***Misleading discussion on batch sizes***
> >
> > Thank you for pointing this out. Our claim that large batch sizes do not help in non-private GAN training is wrong, and we’ve corrected it. Indeed, as pointed out by the reviewer: BigGAN (Brock et al., 2019) uses large batch sizes, and two recent works scaling up StyleGAN to more complex datasets, GigaGAN (Jang et al., 2023) and StyleGAN-XL (Sauer et al., 2023), find benefits in scaling up batch sizes to 1024 and 2048 respectively for complex datasets.
> >
> > The explanation of the noise increasing the complexity of the dataset, necessitating larger batch sizes is interesting, and deserves further investigation.
> >
> > However, we note that in our results, increases in batch size are accompanied by increases in the noise level $\sigma$ – large batches do not simply reduce the variance of the gradient estimate. Moreover, any explanation that does not take into account the privacy budget (which determines how many gradient queries we are allowed) cannot explain improved performance. Phrased alternatively, an explanation that large batch sizes increase performance due to higher quality updates must argue that large batch sizes deliver higher quality updates *per unit of privacy budget*.
> >
> > ***Requested change: writing; suggestions for clarity in abstract and contributions***
> >
> > We’ve modified the abstract to take into account suggestions regarding clarity. We attempt to strike a balance between brevity and technicality (which we try to defer to the manuscript). We’ve also modified Section 1 to take into account suggestions. In particular, we included an extended discussion on balance between discriminator and generators in GAN training. We also made clarifications to the contributions section.
> >
> > ***Requested change: extra seeds***
> >
> > In Figures 1 and 2, and Tables 1, 2, and 3, we run 3 seeds and report the average. In the tables we report std; in the figures we plot the min and max values.
> >
> > ***Requested change: plotting wall clock time***
> >
> > Tables 6 and 7 in Appendix E.1 record the wall clock training times of different settings. Note that since a DPSGD hyperparameter setting and target privacy budget fixes the number of discriminator gradient queries we can make, the effect of increasing $n_\mathcal{D}$ in our experiments is to update the generator less frequently. Hence the $n_\mathcal{D}$=1 setting in Section 4 uses more computation (11h 03m clock time) than the $n_\mathcal{D}=50$ setting (5h 56m). Both are much slower than non-private training (44m).
> >
> > Indeed, the best DP results tend to come from training long with large noise levels, trading off computation for utility (De et al., 2022). For example, the best DP diffusion models (Dockhorn et al., 2022) use 8 V100’s for 1 day to train their best MNIST models. With the caveat that they are not directly comparable, since there are differences in the setup (although A40s are roughly as fast as V100s), our best $\varepsilon = 10$ results train in 7.5 hours on 1 A40.
> >
> > ***Regarding FID as a measure of utility***
> >
> > In terms of utility: if one believes FID is a good measure of the quality of generated data in the non-private case, then is it also a good measure of the quality of generated data in the private case.
> >
> > We provide the clarification that since FID is a distributional metric, there are no inherent limits to the FID that can be achieved under DP. To see this, consider a setting where we are given arbitrarily large amounts of training data. We can run DPSGD with arbitrarily low noise, mirroring non-private training and still obtain our desired DP guarantee.
> >
> > Phrased another way, low FID does not rule out privacy. It may be the case however that low FID is sufficient but not necessary for high “quality” private synthetic data, however deciding on the right definition here is very much an open problem.
> >
> > ***Requested change: Clarifying motivation***
> >
> > Our overall motivation is to demonstrate that DPGANs perform better than previously reported, and explain some important considerations when privatizing GAN training recipes. It seems that the sentence was cut-off or otherwise left incomplete. If the reviewer could finish this sentence, we'd be happy to discuss this point further.
> >
> > ***Have you tried increasing the step size for the discriminator?***
> >
> > In Appendix E.2, we report some results for increasing step size instead of $n_\mathcal{D}$, targeting $\varepsilon =10$ on MNIST. Our preliminary investigations suggest that it does not help as much as taking more steps.
> >
> > ***What would be a recommended way to select $n_\mathcal{D}$ after this study?***
> >
> > We would recommend not to select it directly, but rather use the adaptive discriminator update frequency proposed in Section 6.2.
> >
> > —--------------------------------
> >
> > We thank the reviewer for their suggestions and for engaging with our work. We hope that our comments here and our revisions helped address the reviewer’s concerns.

---

> > > ### Author Response · Authors · 2023-07-06
> > > **References**
> > >
> > > **References**
> > >
> > > Ian Goodfellow, Jean Pouget-Abadie, Mehdi Mirza, Bing Xu, David Warde-Farley, Sherjil Ozair, Aaron Courville, and Yoshua Bengio. Generative adversarial nets. In *Advances in Neural Information Processing Systems 27 (NIPS’14)*, 2014.
> > >
> > > Takeru Miyato, Toshiki Kataoka, Masanori Koyama, and Yuichi Yoshida. Spectral normalization for generative adversarial networks. In *6th International Conference on Learning Representations (ICLR 2018)*, 2018.
> > >
> > > Gauthier Gidel, Hugo Berard, Gaëtan Vignoud, Pascal Vincent, and Simon Lacoste-Julien. A variational inequality perspective on generative adversarial networks. In *7th International Conference on Learning Representations (ICLR 2019)*, 2019.
> > >
> > > Andrew Brock, Jeff Donahue, and Karen Simonyan. Large scale GAN training for high fidelity natural image synthesis. In *7th International Conference on Learning Representations (ICLR 2019)*, 2019.
> > >
> > > Martin Arjovsky, Soumith Chintala, and Léon Bottou. Wasserstein generative adversarial networks. In *Proceedings of the 34th International Conference on Machine Learning (ICML’17)*, 2017.
> > >
> > > Tero Karras, Samuli Laine, and Timo Aila. A style-based generator architecture for generative adversarial networks. In *Proceedings of the 2019 IEEE Conference on Computer Vision and Pattern Recognition (CVPR’19)*, 2019.
> > >
> > > Tero Karras, Samuli Laine, Miika Aittala, Janne Hellsten, Jaakko Lehtinen, and Timo Aila. Analyzing and improving the image quality of stylegan. In *Proceedings of the 2020 IEEE Conference on Computer Vision and Pattern Recognition (CVPR’20)*, 2020.
> > >
> > > Tero Karras, Miika Aittala, Samuli Laine, Erik Härkönen, Janne Hellsten, Jaakko Lehtinen, and Timo Aila. Alias-free generative adversarial networks. In *Advances in Neural Information Processing Systems 34 (NeurIPS’21)*, 2021.
> > >
> > > Axel Sauer, Katja Schwarz, and Andreas Geiger. StyleGAN-XL: Scaling styleGAN to large diverse datasets. In *Special Interest Group on Computer Graphics and Interactive Techniques Conference (SIGGRAPH’22)*, 2022.
> > >
> > > Liyang Xie, Kaixiang Lin, Shu Wang, Fei Wang, and Jiayu Zhou. Differentially private generative adversarial network. *CoRR*, abs/1802.06739, 2018.
> > >
> > > Reihaneh Torkzadehmahani, Peter Kairouz, and Benedict Paten. DP-CGAN: Differentially private synthetic data and label generation. In *IEEE Conference on Computer Vision and Pattern Recognition Workshops, (CVPR Workshops’19)*, 2019.
> > >
> > > Soumith Chintala, Emily Denton, Martin Arjovsky, and Michael Mathieu. How to train a GAN? Tips and tricks to make GANs work. GitHub, 2016.
> > >
> > > Yunhui Long, Boxin Wang, Zhuolin Yang, Bhavya Kailkhura, Aston Zhang, Carl Gunter, and Bo Li. G-PATE: Scalable differentially private data generator via private aggregation of teacher discriminators. In *Advances in Neural Information Processing Systems 34 (NeurIPS’21)*, 2021.
> > >
> > > Dingfan Chen, Tribhuvanesh Orekondy, and Mario Fritz. GS-WGAN: A gradient-sanitized approach for learning differentially private generators. In *Advances in Neural Information Processing Systems 33 (NeurIPS’20)*, 2020.
> > >
> > > Tianshi Cao, Alex Bie, Arash Vahdat, Sanja Fidler, and Karsten Kreis. Don’t generate me: Training differentially private generative models with Sinkhorn divergence. In *Advances in Neural Information Processing Systems 34 (NeurIPS’21)*, 2021.
> > >
> > > Margarita Vinaroz, Mohammad-Amin Charusaie, Frederik Harder, Kamil Adamczewski, and Mi Jung Park. Hermite polynomial features for private data generation. In *Proceedings of the 39th International Conference on Machine Learning (ICML’22)*, 2022.
> > >
> > > Boxin Wang, Fan Wu, Yunhui Long, Luka Rimanic, Ce Zhang, and Bo Li. DataLens: Scalable privacy preserving training via gradient compression and aggregation. In *CCS’21: 2021 ACM SIGSAC Conference on Computer and Communications Security*, 2021.
> > >
> > > Samy Jelassi, David Dobre, Arthur Mensch, Yuanzhi Li, and Gauthier Gidel. Dissecting adaptive methods in GANs. *CoRR*, abs/2210.04319, 2022.
> > >
> > > Tanner Fiez and Lillian J. Ratliff. Local convergence analysis of gradient descent ascent with finite timescale separation. In *9th International Conference on Learning Representations (ICLR 2021)*, 2021
> > >
> > > Soham De, Leonard Berrada, Jamie Hayes, Samuel L Smith, and Borja Balle. Unlocking high-accuracy differentially private image classification through scale. *CoRR*, abs/2204.13650, 2022.
> > >
> > > Tim Dockhorn, Tianshi Cao, Arash Vahdat, and Karsten Kreis. Differentially private diffusion models. *CoRR*, abs/2210.09929, 2022.

---

### Author Response · Authors · 2023-06-26
**Update**

Dear reviewers,

Thank you for your patience. To keep you in the loop, we are still preparing a revision in line with the thoughtful and valuable feedback provided. This has been slower than expected due to some loss of computing resources since the initial submission, and some other circumstances. We expect to provide a revised version by this Friday, June 30, as well as responses to the reviewer questions.

Thanks,
The authors of PG,R.

---

> ### Author Response · Authors · 2023-07-01
> **Update'**
>
> Apologies, but the aforementioned circumstances have caused things to take slightly more time than anticipated. All the experiments we intended to run are now complete, and we should have a revision and responses to all the reviewer questions over the weekend.
>
> Thanks once more for your patience,
> The authors of PG,R.

---

### Author Response · Authors · 2023-07-06
**Thank you**

Thank you for all the helpful suggestions and insightful comments regarding our work! Thank you for your patience in waiting for us to run extra experiments, report new results, and revise our draft – we think the paper is much improved as a consequence. We apologize for the delay in our response.

We’ll highlight a couple of the main changes:

**Experiments:**
- We ran the main experimental results (Figures 1 and 2, Tables 1, 2, 3) with 3 seeds, reporting standard deviation and ranges. The results are robust.
- In Appendix D, we carry out ablation studies where we investigate the phenomenon of taking more steps under various changes to the setup: discriminator sizes, learning rates, batch sizes and noise level. More steps generally help, but in some cases, less so.

**Claims:**
- We’ve restricted our scope to image generation, not synthetic data generation in general, since we do not have experiments for other modalities.
- We’ve corrected the claim that large batch sizes do not help in the non-private setting.

**Writing:**
- We've clarified the relationship to prior work on DPGANs (Section 2), and added more discussion of the use of multiple steps in the non-private GAN literature (Section F).
- We've extended the discussion on mode collapse.
- We've added more detail to the abstract and contributions section.

To conclude, we’d like to reiterate what we believe is the core contribution of our work: falsifying the predominant view in the literature (corroborated by numerous studies presented at top venues: NeurIPS (Chen et. al, 2020, Long et al., 2021, Cao et al., 2021), ICLR (Jordon et al., 2019), ICML (Vinaroz et al., 2022), AISTATS (Harder et al., 2021), CCS (Wang et al., 2021)) that DPSGD-trained GANs are ineffective, and are dramatically outperformed by various custom approaches designed with privacy in mind (see in Table 2, the large performance gaps between previously reported DPGAN results and the various custom approaches). Our work challenges conventional wisdom regarding how to create private generative models, pointing toward a new direction for the area.

—---------------------------

Thank you again for all your comments! We hope our responses and revisions address the concerns raised by the reviewers.

---

> ### Author Response · Authors · 2023-07-06
> **References**
>
> **References**
>
> Dingfan Chen, Tribhuvanesh Orekondy, and Mario Fritz. GS-WGAN: A gradient-sanitized approach for learning differentially private generators. In *Advances in Neural Information Processing Systems 33 (NeurIPS’20)*, 2020.
>
> Yunhui Long, Boxin Wang, Zhuolin Yang, Bhavya Kailkhura, Aston Zhang, Carl Gunter, and Bo Li. G-PATE: Scalable differentially private data generator via private aggregation of teacher discriminators. In *Advances in Neural Information Processing Systems 34 (NeurIPS’21)*, 2021.
>
> Tianshi Cao, Alex Bie, Arash Vahdat, Sanja Fidler, and Karsten Kreis. Don’t generate me: Training differentially private generative models with Sinkhorn divergence. In *Advances in Neural Information Processing Systems 34 (NeurIPS’21)*, 2021.
>
> James Jordon, Jinsung Yoon, and Mihaela van der Schaar. PATE-GAN: Generating synthetic data with differential privacy guarantees. In *7th International Conference on Learning Representations (ICLR’19)*, 2019.
>
> Margarita Vinaroz, Mohammad-Amin Charusaie, Frederik Harder, Kamil Adamczewski, and Mi Jung Park. Hermite polynomial features for private data generation. In *Proceedings of the 39th International Conference on Machine Learning (ICML’22)*, 2022.
>
> Frederik Harder, Kamil Adamczewski, and Mijung Park. DP-MERF: Differentially private mean embeddings with random features for practical privacy-preserving data generation. In *24th International Conference on Artificial Intelligence and Statistics (AISTATS’21)*, 2021.
>
> Boxin Wang, Fan Wu, Yunhui Long, Luka Rimanic, Ce Zhang, and Bo Li. DataLens: Scalable privacy preserving training via gradient compression and aggregation. In *CCS’21: 2021 ACM SIGSAC Conference on Computer and Communications Security*, 2021.

---

### Decision · Action_Editors · 2023-08-25

**Recommendation:** Accept as is

**Comment:**

Reviewers initially found that the significance of the work is low,  as it is just about one hyperparameter whose effects were previously well-known.  The authors did a good job discussing the non-private GAN work, and clearly discussed their contribution in a revision.  Other reviewers had detailed pointers on improving the experiments and presentation. After (essentially) a major revision, the quality of the paper seems to have improved a lot.  Reviewers are now leaning towards accepting the paper.

I agree with the reviewers.  I recommended "accept as is" but since Reviewer UGFA did not have time to respond, the authors should do their best to go over the comments of UGFA again and make sure that every point in the review is satisfactorily addressed in the rebuttal.

Other comments

-  Tuning one more hyperparameter clearly helps, but I feel that it is time for the empirical DP community to start doing honest hyperparameter tuning with DP.   It is fine to argue that existing work does not report such results, but the problem and the rationale behind the choice to continue this practice should be explained in the paper.

- Please align the color choices for different n_D parameters in Figure 2, e.g., Blue is n_D = 1 in (a) but 10 in (b) and (c), which may cause confusion.

**Audience:**

GAN has a large audience and a differentially private version of GAN is a popular baseline for DP synthetic image data generation. People who work on these problems are clearly interested in seeing how the hyperparameter $n_D$ should be chosen.

Diffusion models are obviously more popular than GANs these days for generative tasks, but the performance comparison under DP constraints is actually not clear. I believe this paper provides a better baseline for DP-GAN for future work that aims at evaluating DP-Diffusion models.

**Claims And Evidence:**

The paper considers private GAN with DP-SGD. The main claim of this paper is empirical --- it is important to increase the number of updates to the discriminator per each generator update.  While this was a known technique, the paper is the first to point this out and systematically study its effect under differential privacy constraints.

The support for the claim is empirical. The experiments, in my opinion, provide sufficient evidence for the main thesis of the work.